# Novel Transcriptional and Translational Biomarkers of Tularemia Vaccine Efficacy in a Mouse Inhalation Model: Proof of Concept

**DOI:** 10.3390/microorganisms10010036

**Published:** 2021-12-26

**Authors:** Qing Yan Liu, Sonia Leclerc, Youlian Pan, Ziying Liu, Felicity Stark, Joseph Wayne Conlan

**Affiliations:** 1Human Health Therapeutics Research Centre, National Research Council of Canada (NRC), Ottawa, ON K1A 0R6, Canada; Qing.liu@nrc-cnrc.gc.ca (Q.Y.L.); Sonial.Leclerc@nrc-cnrc.gc.ca (S.L.); felictiy.stark@nrc-cnrc.gc.ca (F.S.); 2Digital Technologies Research Centre, National Research Council of Canada (NRC), Ottawa, ON K1A 0R6, Canada; youlian.pan@nrc-cnrc.gc.ca (Y.P.); Ziying.liu@nrc-cnrc.gc.ca (Z.L.)

**Keywords:** tularemia, *Francisella tularensis*, live attenuated vaccine, transcriptomics, correlates of protection

## Abstract

*Francisella tularensis* subspecies *tularensis* (*Ftt)* is extremely virulent for humans when inhaled as a small particle aerosol (<5 µm). Inhalation of ≥20 viable bacteria is sufficient to initiate infection with a mortality rate ≥30%. Consequently, in the past, *Ftt* became a primary candidate for biological weapons development. To counter this threat, the USA developed a live vaccine strain (LVS), that showed efficacy in humans against inhalation of virulent *Ftt*. However, the breakthrough dose was fairly low, and protection waned with time. These weaknesses triggered extensive research for better vaccine candidates. Previously, we showed that deleting the *clpB* gene from virulent *Ftt* strain, SCHU S4, resulted in a mutant that was significantly less virulent than LVS for mice, yet better protected them from aerosol challenge with wild-type SCHU S4. To date, comprehensive searches for correlates of protection for SCHU S4 Δ*clpB* among molecules that are critical signatures of cell-mediated immunity, have yielded little reward. In this study we used transcriptomics analysis to expand the potential range of molecular correlates of protection induced by vaccination with SCHU S4 Δ*clpB* beyond the usual candidates. The results provide proof-of-concept that unusual host responses to vaccination can potentially serve as novel efficacy biomarkers for new tularemia vaccines.

## 1. Introduction

*Francisella tularensis* subspecies *holarctica* (*Fth*) and subspecies *tularensis* (*Ftt*) are zoonotic facultative intracellular bacterial pathogens, capable of causing a spectrum of diseases collectively called tularemia (reviewed in [1]). Both subspecies can cause serious infections in humans dependent on their portal of entry into the host. *Ftt* is particularly lethal for humans when inhaled as a small particle (<5 µm) aerosol. In this situation as few as 20 inhaled colony forming units (CFU) of *Ftt* can cause systemic potentially lethal infection (≥30% mortality without effective treatment) [2,3,4,5]. In contrast, [6,7], *Fth* rarely results in death regardless of how it enters the host [8,9].

The high mortality associated with inhalation of low doses of *Ftt* made it a major focus of biological warfare programs during the last century [10,11,12,13,14]. To counter this threat, US scientists obtained a live attenuated *Fth* vaccine strain, strain S15, from Russia from which they derived what became known as *Fth* live vaccine strain (LVS) [15]. Its efficacy following scarification, aerosol, or oral administration was demonstrated in human volunteers in the early 1960s [16,17,18,19] and in field trials on tularemia researchers [20]. Overall, LVS given by scarification was particularly effective against subsequent intradermal (ID) infection with virulent *Ftt* strain, SCHU S4, but appeared suboptimal against aerosol challenge. Against the latter, aerosol immunization was more effective, but caused mild to moderate tularemia when administered at the most efficacious doses [19,21]. Consequently, scarification is the sole administration route recommended for humans. However, LVS remains unlicensed and is unavailable for general use.

The emerging threat of bioterrorism at the beginning of this century, triggered by the dissemination of anthrax spores through the US mail system, led to renewed interest in developing countermeasures against potential bioweapons in general, including vaccines against *Ftt* [22]. Our approach to the latter was to make gene deletion mutants of SCHU S4 and to test any strains that were at least as attenuated as LVS for their ability to protect mice from either ID or respiratory challenge with *virulent Ftt* [23,24,25,26,27]. Only one out of sixty mutants tested fulfilled our criteria; a mutant, SCHU S4 Δ*clpB*, from which the chaperonin gene, *clpB*, was deleted [24,25,26,27,28]. Given intranasally (IN) to mice, SCHU S4Δ*clpB* (hereafter Δ*clpB*) was less virulent than LVS, and administered ID was more efficacious against aerosol or intranasal (IN) infection with fully virulent *Ftt*. We also generated other highly attenuated mutants with lesser degrees of efficacy than Δ*clpB* or LVS [24,25,27]. For both respiratory and ID challenge only vaccination with Δ*clpB* proved to be superior to LVS. The reason for this superior protection remains unknown, despite concerted efforts to define differences in the host molecular immune response to vaccination with Δ*clpB* vs. LVS and other attenuated strains of varying efficacy [24,26,27,29].

The literature overwhelmingly shows that canonical cell-mediated immune (CMI) responses rather than antibody responses to vaccination with LVS account for its protective capabilities (reviewed in [30,31]). However, no vaccines currently in clinical use have ever been approved based on the CMI responses they evoke, even when this is the presumed mechanism of action. Another major developmental hurdle for tularemia vaccines is the dearth of natural respiratory infections with *Ftt* that precludes the usual use of large-scale phase 3 clinical trials to determine their efficacy in humans. Instead, the US FDA has developed a policy known as “The Animal Rule” to enable licensing of countermeasures against *Ftt* and other potential biological weapons [32,33]. Specifically, this regulatory pathway allows for evaluation of novel vaccine efficacy using appropriate animal models of infection that lend themselves to a rational means to bridge their correlates of protection (CoP) to human immune responses to vaccination. With these issues in mind others have used a variety of CMI- and antibody- based assays using material obtained from various hosts, including humans, immunized with LVS in search of putative pan-specific immune CoP [34,35,36,37,38,39,40,41,42,43,44,45].

In contrast, we have compared antibody and CMI responses in mice immunized ID with experimental vaccine strains of varying efficacy in BALB/c or C57BL/6 mice. These include extensive immunoproteomic studies to determine the antibody repertoires elicited by these experimental vaccines and kinetics of production of selected cytokines and chemokines in the skin, lungs, livers, spleens, and blood of mice at various times following vaccination or challenge. These have essentially left us empty handed save for the fact that using three distinct vaccination regimens, protection was associated with elevated pulmonary IL-17 levels on day 7 after IN challenge with SCHU S4. However, for a fast-acting pathogen such as *F. tularensis*, CoP need to be detectable as early as possible after vaccination rather than after challenge. In this regard, the multiplex assays for cytokines and chemokines are limited by the relatively small range of antibodies available that are primarily aimed at detecting canonical immune responses, whereas recent transcriptomics and other molecular immunological approaches have shown that non canonical host responses can predict protective responses elicited by vaccines against several other pathogens and LVS in experimental animals and humans [34,35,37,40,41,42,43,44]. Therefore, we were interested to see whether a transcriptomics approach bolstered by a concomitant change in the level of selected associated proteins would reveal unique and robust CoP against respiratory challenge with SCHU S4 induced by immunization with Δ*clpB*.

## 2. Methods

### 2.1. Bacteria

The SCHU S4 mutants were generated as previously described [23] and their safety and efficacy characteristics are summarized in Table 1 along with those of LVS. SCHU S4 is a virulent *Ftt* strain with an LD_50_ for mice of <10 CFU by ID, IN, and aerosol routes of challenge and has been described by us previously [46].

### 2.2. Vaccination of Mice

Young adult female BALB/c mice (*n* = 4/group) were immunized ID with 10^5^ CFU of one or other of the vaccine strains listed in Table 1. Immunization was performed by inoculation of 50 µL of bacteria at a concentration of ~2 × 10^6^ CFU/mL into the shaved mid-belly. The formation of an overt bleb at the site of inoculation was deemed to be indicative of successful ID administration. Four days after vaccination, mice were killed and serum was prepared from whole blood, and spleens were removed intact. Untreated mice were used as negative (naïve) controls. This work was performed under National Research Council Canada animal use protocol # 2015.01 in accordance with the Canadian Council on Animal Care Guidelines for the use and care of laboratory animals (https://ccac.ca/en/standards/guidelines/; accessed on 6 July 2021). For IN challenges, 10 µL of inoculum was added to each nostril of mice whilst under general anaesthesia followed by 10 µL of saline to chase the challenge inoculum into the lower airways.

### 2.3. Transcriptomics

Total RNA was isolated from the spleens of mice vaccinated ID with LVS or one of the SCHU S4 deletion mutants Δ*clpB*, Δ*gplX* and Δ*lpcC*, (4 spleens from each group treated individually throughout) using Tri reagent (Life Technologies, Carlsbad, CA, USA). Genomic DNA contamination was removed by Turbo DNA-Free Kit (Life Technologies). RNA quality was assessed using Agilent Bioanalyzer 2100. RNA-Seq Libraries were generated using the TruSeq strand RNA kit (Illumina, San Diego, CA, USA). The RNA-Seq libraries were quantified by Qbit and qPCR according to the Illumina Sequencing Library qPCR Quantification Guide and the quality of the libraries was evaluated on Agilent Bioanalyzer 2100 using the Agilent DNA-100 chip. The RNA-Seq library sequencing was performed using Illumina Hi-Seq2000 (Genome Quebec, Montreal, QC, Canada). RNA-seq data are available in the GEO repository with access number GSE186408. STAR (v2.7.8a) [48] was used for alignment of the reads to the reference genome and to generate gene-level read counts. Mouse (*Mus musculus*) reference genome (version GRCm39 Gencode M26) [49] and corresponding annotation were obtained from Gencode (https://www.gencodegenes.org/mouse/stats.html (accessed on 2 February 21) and used as reference for RNA-seq data alignment process. DESeq2 [50] was used for data normalization and differentially expressed gene identification for each treatment vs. naïve samples. The expression value of each gene was expressed as average read counts. Differentially expressed genes (DEGs) were obtained by comparing treated samples with naïve samples (control) and all vaccinated samples compared with each other. A *q*-value (adjusted *p*-value) of less than 0.05 and 2 fold change in ratio (abs (log2 fold-change) ≥ 1) were used to generate a DEGs list. KEGG pathway enrichment analyses were done using GOAL software; pathway enrichment *p*-values were computed using the Fisher’s exact test via the hypergeometric distribution and were BH corrected [51].

### 2.4. Multiplex and ELISA Assays

A commercial ELISA kit (My BioSource Inc., San Diego, CA, USA) was used to determine relative levels of Saa3 in mouse sera in accordance with the manufacturer’s instructions. Sera were tested at 1:2000 and 1:10,000 dilutions. Serum levels of tissue inhibitor of metalloprotease 1 (TIMP1), granzyme B, matrix metalloproteinases 3 and 8 (MMP3/8) were determined by Luminex using immunomagnetic multiplex kits (MilliporeSigma, Oakville, ON, Canada). Data were analysed using Kruskal-Wallis test followed by Dunn’s post-test for multiple comparisons. Adjusted *p* values of <0.05 were considered to be statistically significant.

## 3. Results

### 3.1. Transcriptomic Analysis

For reasons of cost and data handling logistics, we chose to examine the transcriptome in the spleens of BALB/c mice four days after ID immunization with one or other of the strains of *F. tularensis* listed in column 1 in Table 1. The spleen was chosen as a substitute for PBMC which are in short supply from individual mice, and day 4 was chosen because that was the time when most splenic cytokine and chemokine levels peaked in our earlier studies using multiplex analysis [27].

On average, 85% of the 34 million paired-end reads in each sample were mapped to the mouse genome. A total of 5361 differentially expressed genes (DEGs) were collectively identified from the 4 pairwise comparisons between the vaccines and the naïve control (Figure 1, Table 2 and Appendix A). Compared to spleens from naïve mice, 3539, 3242, 2006 and 1350 genes were differentially expressed after vaccination with Δ*clpB*, LVS, Δ*gplX* and Δ*lpcC*, respectively. The number of changed genes reflects the extent of host response to vaccination and appears to correlate with the efficacy of the vaccine strains, with Δ*clpB* > LVS > Δ*gplX* > Δ*lpcC*.

Cluster and heatmap analyses of the 5361 genes showed distinct patterns of gene expression for each vaccine strain (Figure 1). Mice immunized with Δ*clpB* and LVS formed one branch, while the other two vaccines and naïve mice formed another branch. Thus, immune responses to Δ*gplX* and Δ*lpcC* are more similar to naïve mice than to mice immunized with Δ*clpB* or LVS. Although the overall host responses to Δ*clpB* and LVS are closely related, there were 139 DEGs when we did pairwise comparisons between these two strains (Table 2). Likewise, the patterns produced by Δ*gplX* and Δ*lpcC* were similar to each other with 397 DEGs between this pair (Table 2).

The geneID, normalized mean read counts and log2 ratio of these changed genes are listed in Appendix A. Reassuringly, Il-6, IFNγ, and Il-17 transcripts were among the top twenty that were significantly overexpressed in mice immunized with Δ*clpB versus* the other SCHU S4 mutants as this is in keeping with our previous findings examining the relative levels of these proteins in the spleens of similarly vaccinated mice [27]. Additionally, upregulation of Il-1α, Il-1β, Cxcl1 and ccl2 (MCP-1) transcripts, though lower down the ranking, also concurred with our prior multiplex studies. They essentially followed the pattern Δ*clpB > LVS >* Δ*gplX >* Δ*lpcC.* This is in overall agreement with the relative protection these strains administered ID provide against respiratory infection with SCHU S4 (Table 1).

Figure 2 illustrates the number of genes identified to be changed uniquely in one or simultaneously in two or more samples (up in Figure 2a, down in Figure 2b). As expected, Δ*clpB* had the highest number of uniquely differentially expressed genes (499 up, 554 down). Δ*clpB* and LVS clearly shared the highest number of up- and down- regulated genes, 852 and 1576 respectively, since they both protect against respiratory challenge, albeit to different extents. A large number of up- (290) and down- (675) regulated genes were shared by Δ*clpB*, LVS and Δ*gplX* as these vaccines all protect against intradermal challenge. While Δ*lpcC* shares some (520 up, 234 down) of the DEGs with Δ*gplX*, it shared very few DEGs with Δ*clpB* and LVS, individually, or with both. There are 56 and 166 commonly up- or down- regulated genes in all 4 samples; they are likely genes responding to general vaccination regardless of the mutant strain used (Appendix A).

Further study of KEGG pathway enrichment analyses of the differentially expressed genes in the three strains that conveyed some levels of protection against challenges revealed several pathways that are expected to be up-regulated after vaccination (Table 3). The most significant pathway is the cytokine–cytokine receptor interaction pathway, followed by NOD-like receptor signaling pathway (families of pattern recognition receptors responsible for detecting various pathogens and generating innate immune responses), chemokine signaling pathway and antigen processing and presentation, as well as IL-17, TNF signaling and viral protein interaction with cytokine and cytokine receptor pathways. For Δ*clpB*, in addition to the genes that were shared with the LVS and Δ*gplX*, there were many more genes that were up-regulated in these pathways. No direct link of the down-regulated pathways can be made to CMI. The neuroactive ligand–receptor interaction pathway that participates in environmental information processing was significantly down-regulated in these three strains. The down-regulation of calcium signaling pathway in all strains may be related to depressed control of fast cellular processes. A large number of genes (46) in the metabolic pathways were uniquely down-regulated in Δ*clpB* (Table 4). Interestingly, 21 genes in the aforementioned pathway were significantly up-regulated in Δ*lpcC* (Appendix A), indicating opposite metabolic process effects of these two vaccines.

Because biomarkers for vaccine strains that outperform LVS need to be robust, we have selected potential transcriptional changes (Table 5) that have to meet the following filtering criteria: (1) the transcripts are highly abundant (read count > 300, in up-regulated testing strain, or in naïve for down-regulation); (2) more than 4-fold changes (|log2FC| > 2) in Δ*clpB* versus naïve; (3) the gene products are known to be expressed in whole blood, either naturally or by secretion; (4) more than two fold changes in Δ*clpB* versus LVS, which could be sufficient to distinguish host responses to these functionally closely related vaccines. By these criteria, some of the genes ranked highly in Appendix A, failed to make the grade for inclusion in Table 5. In addition to their ability to distinguish Δ*clpB* from the others three test vaccines, a majority of these selected genes can be used to separate LVS from Δ*gplX* and Δ*lpcC* and some of them can also be used to distinguish Δ*gplX* from Δ*lpcC* (Appendix A). Δ*lpcC* was unable to protect against either respiratory or intradermal challenge route. Therefore, these down-regulated genes could also be developed as potential indicators of non-protective vaccines. In this regard, all the selected biomarkers down-regulated in Δ*clpB* (1300017J02Rik, Slc6a9, Art4, Sptb, and Aqp1) were significantly up-regulated in Δ*lpcC.* Aqp1, Sptb and Slc6a9 were up-regulated in both Δ*gplX* and Δ*lpcC* which means they can be developed to distinguish between strains with at least some protective activity against respiratory challenge.

### 3.2. Proteomic Confirmation of Transcriptomics Findings

Previously, we have reported [27] that IFNγ, IL-6, CcL2 (MCP-1), and Cxcl1 (KC) proteins are over produced in the spleens and sera of mice immunized ID four days earlier with Δ*clpB*, versus *gplX* or *lpcC*. LVS elicited similar responses to Δ*clpB* (unpublished). In this study, we found that IFNγ, IL-6, and CcL2 are among the up-regulated genes by both Δ*clpB* and LVS. In addition, Saa3 is highly up-regulated by vaccination with both Δ*clpB* (122-fold) and LVS (21-fold). To determine whether these finding hold at the translational level, we first used a commercial ELISA kit to examine serum levels of Saa3 in the same mice that provided the spleens for transcriptional analyses. In sera diluted 2000-fold, Saa3 levels in mice immunized with Δ*clpB* (adjusted *p* = 0.0008) or LVS (adjusted *p* = 0.03), were significantly higher than background (Figure 3). However, at 1:10,000 dilution serum Saa3 levels were only significantly higher than background (adjusted *p* = 0.01) in mice immunized with Δ*clpB* (Figure 3). Therefore, depending on dilution, serum Saa3 levels 4 days after vaccination can discriminate between vaccines that provide some degree of protection against respiratory challenge versus those that do not or Δ*clpB* versus all three other vaccine candidates.

Next, we looked at serum granzyme B, TIMP1, MMP3 and MMP8 levels on day four after vaccination by multiplex (Luminex) assay (Figure 4). The results show that compared to naïve mouse serum, levels of granzyme B, tissue inhibitor of metalloproteinase-1 (TIMP1), and matrix metalloproteinase-8 (MMP8), but not MMP3, were significantly up-regulated in the sera of mice immunized with either Δ*clpB* or LVS even after correction for multiple comparisons. Serum levels of TIMP-1 and MMP-8 were also significantly higher in mice immunized with Δ*clpB* vs. Δ*lpcC*. However, in no cases were levels of these proteins significantly greater in mice immunized with Δ*clpB* versus LVS or Δ*gplX*. It remains to be determined how much this holds true for the other highly up-regulated and down-regulated genes in Table 5 for which proteomic assays were unavailable.

## 4. Discussion

The FDA Animal Rule for the approval of vaccines against *Ftt* requires evidence that CoP from two animal models likely predict their efficacy in humans. Currently, only LVS has been shown to protect humans against inhalation of virulent *Ftt*. This data stems solely from experiments conducted between 1960–1975 in which volunteers or tularemia researchers immunized by various routes with LVS were subsequently exposed to SCHU S4 [16,17,18,21,52,53]. These studies showed that LVS administered by scarification provided the simplest and safest means of eliciting protection, though it proved sub-optimal against aerosol challenge. Had LVS proved to be 100% effective against the latter, then any signs of vaccine take (e.g., eschar formation at the immunization site, or seroconversion to LVS) could have served as a straightforward CoP. Despite vaccine take being 100% in these experiments, in one pivotal study, 80% of unvaccinated individuals became ill within a few days following inhalation of between 10–50 CFU of SCHU S4, as did 3/18 individuals immunized with LVS [17]. Similarly, LVS elicited protection against aerosol challenge with a breakthrough threshold of ~1000 human infectious doses waned from 100% at 2 months to 25% at 11 months after vaccination [2]. When these human experiments were performed, only relatively crude measures of immunity were available. Namely, seroconversion (bacterial agglutination titer) that proved unreliable as a CoP [54]. Thus, neither vaccine take nor antibody titer predicted longer term protection. Because CMI and its critical role in protection against facultative intracellular bacterial pathogens was in its infancy at this time, no attempts were made to measure such immune responses elicted by LVS as potential CoP.

Nowadays, there is abundant animal data showing aspects of CMI that are crucial to protective immunity following vaccination with LVS (reviewed in [31,55]). However, in the absence of any accompanying human challenge data, it is difficult to predict which of these responses might correlate with long-lasting protection given its short-term zenith to rapid nadir in early challenge studies. For novel experimental vaccines, that have only been shown to be effective in animal models, the bridge to predicting their efficacy in humans remains even more challenging. Nevertheless, animal models can at least provide a starting point. In this regard, we have developed a deletion mutant of SCHU S4, Δ*clpB*, that offers better protection than LVS to BALB/c mice challenged IN or by aerosol with virulent *Ftt* [26,27]. Additionally, Δ*clpB*, LVS and Δ*gplX* all protect mice against ID challenge with virulent *Ftt*, whereas mutant strain Δ*lpcC* signally fails in both regards. Our prior attempts to correlate selected molecular immune responses to vaccination with these mutants were unsuccessful and biased by available reagents [24,25,26,27,56]. To determine whether other early host molecular responses to vaccination could predict the relative efficacy of these experimental vaccine strains, we performed more impartial transcriptomic profiling on the spleens of mice vaccinated ID 4 days earlier with 10^5^ CFU of one or other vaccine strain. The spleen was used as a surrogate for PBMC that are in too short supply in individual mice to allow this type of approach. Our analyses revealed several up- and down- regulated genes associated with canonical CMI pathways that correlated with the superior protective capability of Δ*clpB.* Additionally, these studies identified a large number of other potential CoP. Among the most up-regulated transcripts in the spleens of BALB/c mice immunized with Δ*clpB* versus the other three test strains (Table 5 and Appendix A) were, IFNγ, IL-6, Ccl2 (MCP1) and CxCL1 (KC). Interestingly, we previously showed this to be the case when spleen homogenates and serum from mice immunized four days earlier were examined for the proteins encoded by these genes. Moreover, these proteins were all produced in significantly higher quantities in mice immunized with Δ*clpB* versus Δ*gplX* or Δ*lpcC* [27]. However, there were no significant differences in the levels of these proteins produced in the spleens or sera of mice immunized four days earlier with Δ*clpB* vs. LVS. Our statistical analysis of transcript counts confirmed that IFNγ and IL-6 were not significantly up-regulated in mice immunized with Δ*clpB* vs. LVS. Moreover, these proteins were similarly up-regulated in mice that were protected (BALB/c) or not (C57BL/6) by immunization with Δ*clpB* [26]. Overall, our prior findings indicated that our previous selection of serum or splenic cytokines or chemokines were poor CoP. However, these were restricted by commercial availability of antibodies to target immune molecules. In contrast, transcriptomics allows a much broader and less biased view of host responses to vaccination that can reveal non-canonical responses not normally associated with protective CMI (reviewed in [57]). In this regard, the current study revealed several genes, not routinely associated with protective CMI, that were many-fold up-regulated in mice immunized with Δ*clpB* vs. LVS, Δ*gplX* or Δ*lpcC* (Table 5 and Appendix A). For instance, transcripts for serum amyloid A3 (Saa3) were significantly up-regulated relative to naïve mice by 122-fold versus 21-fold by Δ*clpB* vs. LVS, respectively, but not at all by Δ*gplX* and Δ*lpcC*. Therefore, an ELISA specific for mouse Saa3 was used to examine its levels in the sera from the same mice used for splenic transcriptional analysis. The results (Figure 3) clearly recapitulated the transcriptomics data (Δ*clpB* > LVS > Δ*gplX* > Δ*lpcC*), making Saa3 a promising surrogate CoP. This was the case too for serum granzyme B, TIMP1, and MMP8 measured by Luminex whereas MPP3 showed no up-regulation in protein expression (Figure 4). Several other transcripts were also up-regulated by at least 2-fold in Δ*clpB*- vs. LVS-immunized mice and substantially more so compared to mice immunized with Δ*gplX* or Δ*lpcC*. In all our prior comparative efficacy studies of different vaccine strains, LVS was always the next best performing vaccine after Δ*clpB* at providing protection against respiratory challenge with SCHU S4. Therefore, it is unsurprising that they induce the most similar transcriptional profiles in mice. It is interesting to note too, that down-regulation of certain genes (e.g., *Aqp1*, *Sptb)* also correlate with the ability to elicit protection against respiratory challenge (Appendix A). Whether or not any or a small combination of these differences are sufficient to serve as CoP for ranking the relative efficacy of tularemia vaccines in other mammals including humans remains to be determined.

Although our studies were limited to BALB/c mice, several groups have performed comprehensive transcriptional and translational analysis using PBMC recovered from individual humans for up to 2 weeks following immunization with LVS [42,43,57]. Fuller et al. examined the transcriptomes of PBMC taken from volunteers at -6, and 1, 2, 8, 14 days after immunization with LVS from a batch lot produced for the United States Army Medical Institute for Infectious Diseases (USAMRIID) in the early 1960s [42,43]. Despite the age of this vaccine, others have demonstrated that stored at −80 °C, it elicited similar CMI responses in humans over a test period of >35 years [58]. The former studies showed transcriptional changes primarily in molecular pathways associated with innate immunity. Much more recently, others [57] have comprehensively mapped transcriptional changes in PBMC from human volunteers immunized 1, 2, 7 or 14 days earlier with USAMRIID LVS or a newer batch manufactured in 2007 by the Dynport Vaccine Company under more appropriate GMP conditions [59]. Again, the individual responses were very reproducible. However, in both cases, the fold change in transcript abundance was ≤3-fold in either direction compared with the exponentially higher changes observed in mice spleens in the current study. The latter study showed also that both LVS vaccine lots induced small (<2-fold), but significant changes in a few serum cytokines and chemokines in contrast to the >1000-fold increases we previously found in mice [27]. Finally, a proteomics study on PBMC from LVS vaccines on days 7 and 14 showed ≤2-fold changes in abundance of numerous proteins following vaccination [45]. Because of the massive sizes of the datasets produced from human transcriptomics and proteomics studies, it is impossible to do them full credit herein. Instead, we have produced a Appendix A showing the largest changes over time in transcriptomes and proteomes in human PMBC following immunization with LVS. Surprisingly, there was little overlap between the transcription results of Fuller et al. and those of Natrajan et al., Appendix A). This was also the case with the proteomics study by Chang et al. [45]. Finally, none of these datasets showed much overlap with the mouse data generated in the current study. The reasons for all of the aforementioned differences are likely multifold and include: (1), Different host species naturally react differently at the genetic level to vaccination with live tularemia vaccines; (2), the live vaccine candidates examined in the current study cause systemic infection in mice to varying degrees, thus exponentially amplifying the original antigenic burden; (3), serum cytokine responses in mice are diluted in 5.0 mL of blood versus 5.0 L of blood in humans (a differential that is eliminated using the macrophage killing assay (MKA, described below); (4), ultimately it is the acquired immune responses to vaccination that determine the degree of protective immunity and early transcriptional responses in different host species lead to similar acquired immunological outcomes as suggested by the MKA and other assays; (5), PBMC do not fully reflect immunogenetic changes occurring in the lymph nodes, the primary sites of antigen processing and presentation; (6), the gap between day 2 and 7 data collection points for humans miss information generated in mice on day 4 after vaccination; (7), transcriptional changes that occur before the onset of acquired CMI correlate more with mechanisms of protection rather than CoP. In mice and humans, the first measurable evidence of acquired CMI following vaccination with LVS occurs starting at approximately 2 weeks [60,61].

Besides using early post-vaccination transcriptional analyses that occur at the cusp of the acquired immune response, others have developed functional assays, preferred by regulatory agencies, to examine potential CoP following vaccination with live tularemia vaccines including LVS. The most promising of these is the so called “macrophage killing assay” (MKA) developed by Karen Elkins and colleagues at the US FDA in an attempt to reveal CoP for novel tuberculosis and tularemia vaccines [62,63]. Briefly, the assay involves infecting quiescent, adherent host macrophages contained within wells of tissue culture plates with *Ft*. In this condition, *Ft* will grow exponentially within the macrophages and kill them within 72 h. However, if immune T cells from the same, previously vaccinated, host are overlaid on top of the infected macrophages, then *Ft* multiplication is rapidly curtailed and can be measured as a logarithmic decrease in CFU. Additionally, the transcriptomes, and phenotypes of the T cells and macrophages that remain at the end of the assay can be determined as can the contents of the well supernatants [35,36,37,64]. In this regard, we previously showed that the enhanced efficacy of Δ*clpB* vs. LVS in a murine aerosol challenge model was associated with a concomitant increase in the levels of pulmonary IFNγ, TNFα, and IL-17 seven days after challenge [24,26,29]. Using the MKA and these cytokines individually or in mixtures, we have shown that combining all three molecules results in the most effective killing of SCHU S4 (Appendix A). Thus, overall this assay appears to be capable of fully recapitulating the in vivo protective immune response. Moreover, the MKA can be used with multiple species, to allow for the discovery of pan-species specific CoP. In this regard, the MKA has already been successfully employed in mice, rats, and humans immunized with LVS or Δ*clpB* [37,40,41,62,65].

## 5. Concluding Remarks

This study shows that a potentially simple serum-based test for one or a few molecules could be used to develop a CoP profile for humans vaccinated with Δ*clpB*. In particular, this study shows that vaccination with Δ*clpB* especially induces significant up- or down- regulation of genes hitherto not associated with protective immunity to respiratory challenge with virulent *F. tularensis*. A finding in keeping with several bioinformatics studies that have shown unexpected CoP for LVS and several vaccines against other infectious diseases [57]. We have recently made a batch lot of Δ*clpB* under GMP conditions [66] that will be used to conduct clinical trials sometime in 2022 or 2023. Thereafter, we will be able to directly compare human immune responses to vaccination with Δ*clpB* and LVS that could reveal either a common or unique CoP for these two vaccines.

## Figures and Tables

**Figure 1 microorganisms-10-00036-f001:**
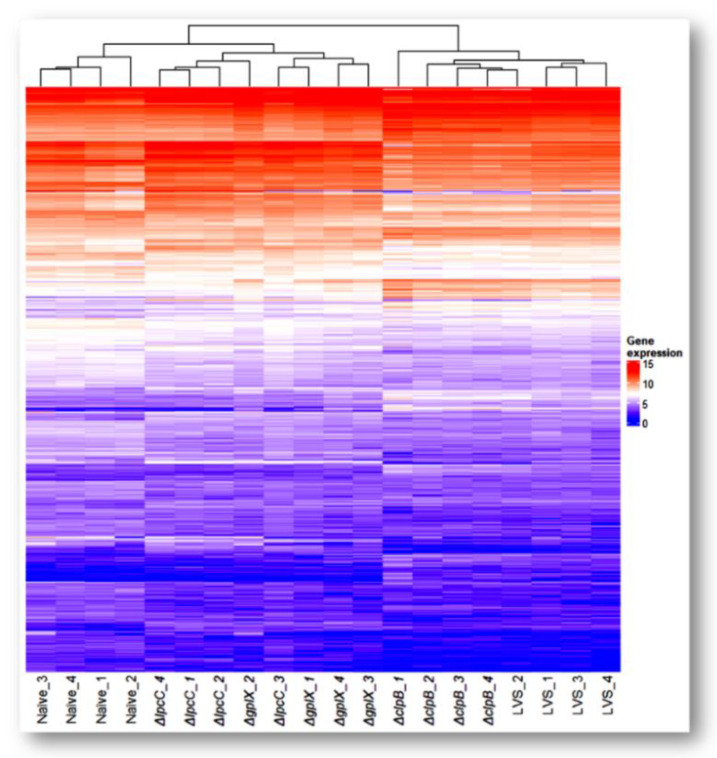
Transcriptome overview. Heatmap of expression profile of differentially expressed genes across four vaccine strains. Genes that changed their expression levels significantly (*p* < 0.05) in at least one of the vaccinated samples when compared with the naïve sample were extracted from the data set. All four replicates of each sample group were included to show reproducibility. A total of 5361 genes were compiled. Data values were log_2_ transformed.

**Figure 2 microorganisms-10-00036-f002:**
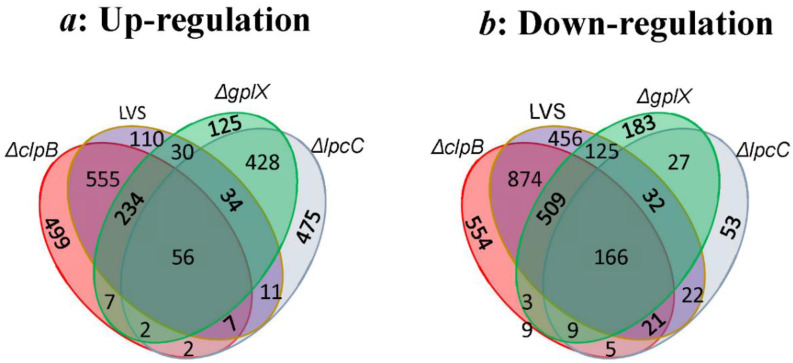
Numbers of differentially expressed genes in all 4 samples illustrated by Venn diagram. All up- (**a**) or down- (**b**) regulated genes, relative to spleens from naïve mice in each sample are encompassed in a colored oval. Shared genes are indicated by numbers situated on appropriate overlapping areas.

**Figure 3 microorganisms-10-00036-f003:**
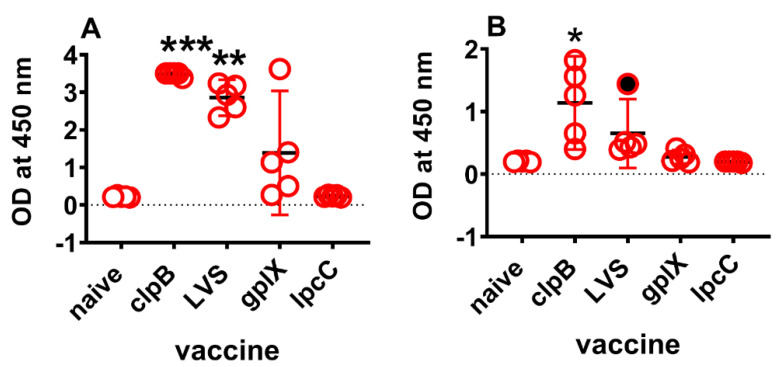
Serum Saa3 levels 4 days after vaccination. Blood was collected from mice (*n* = 5/ group) 4 days after ID vaccination with one or other vaccine strain. Sera were diluted 2000-fold (**A**) or 10,000-fold (**B**) and tested for the presence of Saa3 according to the manufacturer’s instructions. Colour reaction was stopped after 15 minutes. Graphs were plotted as means (horizontal black dash) with 95% CI (red vertical lines). Dotted line is the limit of detection. Asterisks denote significantly higher levels than naïve sera by Kruskal Wallis analysis followed by Dunn’s test for multiple comparisons. *** (adjusted *p* = 0.0008), ** (adjusted *p* = 0.03), * (adjusted *p* = 0.01). Filled circle, outlier identified by ROUT analysis and excluded from calculations.

**Figure 4 microorganisms-10-00036-f004:**
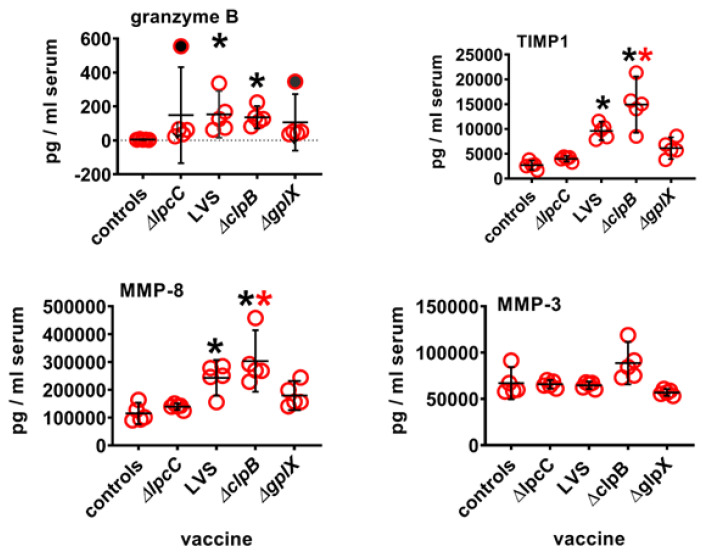
Serum protein levels four days after immunization. Mice (*n* = 4–5/group) were immunized ID with 10^5^ CFU of one or other vaccine strain. Four days later serum was prepared from each mouse and assayed for the presence of Granzyme B, TIMP1, MMP3, and MMP8. Data were analysed using Kruskal Wallis test to compare each group to every other group followed by Dunn’s correction for multiple comparisons. Filled circles (outliers by ROUT analysis) removed prior to calculations. Black asterisks, significantly greater than levels found in control serum, red asterisks significantly greater than Δ*lpcC* (adjusted *p* ≤ 0.04); 95% CI (vertical black lines) and means (horizontal black lines).

**Table 1 microorganisms-10-00036-t001:** Selected characteristics of *F. tularensis* strains used in the current study.

Mutant	IN LD_50_ (CFU)	ID LD_50_ (CFU)	Survival against ID Challenge with SCHU S4 ^a^	Survival against Respiratory Challenge with SCHU S4 ^b^	Ref
LVS	10^3^	>10^7^	100%	0–20%	[24,25]
Δ*clpB*	>10^4^ <10^6 c^	>10^7^	100%	60–100%	[24,27,47]
Δ*gplX*	NT	>10^7^	80%	0%	[27]
Δ*lpcC*	>10^3^ <10^5^	>10^7^	0%	0%	[47]

^a^ Challenge dose ≤ 10^5^ CFU; ^b^ IN or aerosol challenge dose ≤ 200 CFU; ^c^ range from multiple tests.

**Table 2 microorganisms-10-00036-t002:** Number of differentially expressed genes between each comparison.

	Δ*clpB*/ Naïve	LVS/ Naïve	Δ*gplX*/ Naïve	Δ*lpcC*/ Naïve	Δ*clpB/* LVS	Δ*clpB*/ Δ*gplX*	Δ*clpB*/ Δ*lpcC*	LVS/ Δ*gplX*	LVS/ Δ*lpcC*	Δ*gplX*/ Δ*lpcC*
Up	1362	1037	916	1015	108	961	1475	358	897	233
Down	2177	2205	1090	335	31	1219	1926	722	1546	164

**Table 3 microorganisms-10-00036-t003:** Participation of up-regulated genes in KEGG pathways in the three vaccine strains that protect against ID challenge with SCHU S4 challenge.

KEGG Pathway ID	KEGG Pathway Name	Total Known Genes	Δ*clpB* Alone		Δ*clpB* and LVS Shared		Δ*clpB*, LVS and Δ*gplX* Shared		Total Gene Counts ^a^
			padj	Matched Genes	padj	Matched Genes	padj	Matched Genes	
mmu04060	Cytokine–cytokine receptor interaction	292	3.98 × 10^−8^	Ccr1, Il21, Cx3cl1, Cd70, Cxcr1, Il17f, Il23a, Osmr, Tnfrsf8, Tnfsf9, Tnfrsf1b, Ccl11, Ccl22, Ccl9, Ccl6, Fasl, Ccl3, Il23r, Ccr8, Fas, Gdf15, Ltbr, Cxcl16	1.15 × 10^−14^	Il10, Il11, Il33, Il13, Il12b, Il2ra, Ccl17, Il15ra, Tnf, Tnfsf15, Bmp10, Csf2rb2, Ccl8, Tnfrsf1a, Csf2rb, Tnfrsf11b, Ccl4, Il1f9, Cxcl1, Ccr5, Cxcl2, Cxcl5, Il12rb2, Lif, Tnfsf10, Il17a, Ccl24, Il1a, Tnfrsf9, Il1b, Il1r2, Xcl1, Inhba, Tnfrsf12a, Inhbb, Csf2, Csf3	5.75 × 10^−6^	Cxcl11, Il12rb1, Il22, Cxcl10, Il27, Il6, Ccl12, Cxcl9, Ccl7, Ifng, Ccl2, Il1rn, Cxcl3	73
mmu04621	NOD-like receptor signaling pathway	211			3.29 × 10^−5^	Jun, Pycard, Nlrp3, Nod2, Nod1, Mefv, Nlrp1b, Myd88, Tnf, Ifi207, Ifi206, Ifi204, Ripk3, Cybb, Il1b, Cxcl1, Cxcl2, Tlr4	1.16 × 10^−14^	Txn1, Gbp5, Mapk13, Gbp7, Nampt, Nlrp1a, Oas1g, Il6, Ccl12, Gbp2b, Oas2, Oas1a, Irf7, Ccl2, Stat2, Stat1, Gbp2, Cxcl3, Gbp4, Gbp3	38
mmu04062	Chemokine signaling pathway	192	7.03 × 10^−4^	Ccr1, Hck, Ccl11, Ccl22, Ccl9, Cx3cl1, Ccl6, Cxcr1, Ccl3, Ccr8, Cxcl16, Gng12	9.03 × 10^−3^	Ccl24, Ccl8, Ccl4, Gnb4, Stat3, Pik3r6, Xcl1, Cxcl1, Ccl17, Ccr5, Cxcl2, Cxcl5	1.19 × 10^−4^	Cxcl11, Ccl12, Cxcl10, Cxcl9, Ccl7, Ccl2, Stat2, Stat1, Cxcl3	33
mmu04612	Antigen processing and presentation	90	1.06 × 10^−4^	H2-T24, Psme2b, Ctss, H2-K1, H2-Q4, Lgmn, H2-T3, H2-Q1, Hsp90aa1	2.23 × 10^−7^	H2-T23, H2-T10, H2-Q6, H2-Q7, Hspa1b, Tap2, Hspa1a, H2-Q2, Psme1, Tnf, Hspa8, Gm11127, Tapbp, B2m	0.01	H2-T22, Ifng, Tap1, Psme2	27
mmu04514	Cell adhesion molecules	174	7.15 × 10^−5^	H2-T24, Sdc4, H2-K1, Ctla4, H2-Q4, Pdcd1, H2-T3, Tigit, H2-Q1, Nectin2, Selp, Mag, Cldn1	1.44 × 10^−3^	H2-T23, H2-T10, H2-Q6, Sdc3, H2-Q7, H2-Q2, Nrcam, Pdcd1lg2, Itgam, Vcan, Ocln, Gm11127, Icam1			26
mmu04610	Complement and coagulation cascades	93	3.32 × 10^−3^	C1rb, C6, F13a1, Plat, Bdkrb1, Plaur, C2	6.68 × 10^−9^	C1qb, Procr, C1ra, F10, Serping1, C5ar1, C3, F7, C1s2, Itgam, C1s1, Plau, C3ar1, Serpine1, A2m, Cfb			23
mmu04940	Type I diabetes mellitus	70	1.40 × 10^−5^	H2-T24, Fasl, Ptprn, H2-K1, H2-Q4, Fas, Prf1, H2-T3, H2-Q1	4.11 × 10^−6^	H2-T23, Il1a, H2-T10, Gm11127, H2-Q6, Il1b, H2-Q7, Il12b, H2-Q2, Tnf, Hspd1	0.03	H2-T22, Ifng, Gzmb	23
mmu04630	JAK-STAT signaling pathway	168			2.37 × 10^−5^	Il10, Il11, Socs3, Il12rb2, Il13, Il12b, Il2ra, Lif, Il15ra, Csf2rb2, Csf2rb, Myc, Stat3, Cdkn1a, Csf2, Csf3	1.41 × 10^−3^	Il12rb1, Socs1, Il22, Il6, Ifng, Stat2, Stat1	23
mmu04210	Apoptosis	136	1.29 × 10^−4^	Ctss, Ctsd, Gadd45g, Fasl, Ctsz, Gadd45b, Fas, Prf1, Casp12, Bcl2a1b, Bcl2a1a	1.76 × 10^−3^	Ctsc, Csf2rb2, Jun, Tnfrsf1a, Csf2rb, Cycs, Fos, Daxx, Tnf, Tnfsf10, Tuba8			22
mmu04620	Toll-like receptor signaling pathway	100	4.97 × 10^−3^	Ccl3, Tlr8, Tlr7, Cd14, Ikbke, Tlr3, Tlr2	7.77 × 10^−3^	Jun, Il1b, Ccl4, Il12b, Fos, Myd88, Tlr4, Tnf	5.88 × 10^−5^	Cxcl11, Mapk13, Il6, Cxcl10, Cxcl9, Irf7, Stat1	22
mmu05332	Graft-versus-host disease	63	4.67 × 10^−5^	H2-T24, Fasl, H2-K1, H2-Q4, Fas, Prf1, H2-T3, H2-Q1	6.84 × 10^−5^	H2-T23, Il1a, H2-T10, Gm11127, H2-Q6, Il1b, H2-Q7, H2-Q2, Tnf	3.54 × 10^−3^	H2-T22, Il6, Ifng, Gzmb	21
mmu05330	Allograft rejection	63	4.67 × 10^−5^	H2-T24, Fasl, H2-K1, H2-Q4, Fas, Prf1, H2-T3, H2-Q1	6.84 × 10^−5^	Il10, H2-T23, H2-T10, Gm11127, H2-Q6, H2-Q7, Il12b, H2-Q2, Tnf	0.02	H2-T22, Ifng, Gzmb	20
mmu04650	Natural killer cell mediated cytotoxicity	121	9.74 × 10^−4^	Raet1e, Klra7, Fasl, Sh2d1b2, H2-K1, Fas, Prf1, Ulbp1, Lcp2	0.02	H2-T23, Raet1d, Klrk1, Fcer1g, Icam1, Csf2, Tnf, Tnfsf10			17
mmu04640	Hematopoietic cell lineage	94	0.05	Cd1d2, Sco1, Cd38, Cd14, Cd44	7.16 × 10^−5^	Il11, Il1a, Itgam, Il1b, Il1r2, Il2ra, Anpep, Itga5, Csf2, Csf3, Tnf			16
mmu05320	Autoimmune thyroid disease	79	3.77 × 10^−5^	H2-T24, Fasl, H2-K1, Ctla4, H2-Q4, Fas, Prf1, H2-T3, H2-Q1	7.28 × 10^−3^	Il10, H2-T23, H2-T10, Gm11127, H2-Q6, H2-Q7, H2-Q2			16
mmu05416	Viral myocarditis	88	0.01	H2-T24, H2-K1, H2-Q4, Prf1, H2-T3, H2-Q1	3.59 × 10^−3^	H2-T23, H2-T10, Gm11127, Cycs, H2-Q6, H2-Q7, Icam1, H2-Q2			14
mmu00010	Glycolysis/Gluconeogenesis	67			2.64 × 10^−6^	Aldh1b1, Pkm, Tpi1, Ldha, Eno1b, Pgk1, Pgam1, Eno1, Gapdh, Hk2, Pfkp			11
mmu04623	Cytosolic DNA-sensing pathway	63			0.03	Il33, Pycard, Ripk3, Il1b, Ccl4	3.60 × 10^−5^	Il6, Cxcl10, Zbp1, Cgas, Irf7, Ifi202b	11
mmu03050	Proteasome	47			0.01	Pomp, Psmb2, Psme1, Psma8, Psmb8	9.78 × 10^−5^	Psma5, Ifng, Psme2, Psmb10, Psmb9	10
mmu05133	Pertussis	77	0.02	C1rb, Casp1, Cd14, Il23a, C2	1.11 × 10^−16^	C1qb, Il10, C1ra, Jun, Pycard, Serping1, Nos2, Il12b, Itga5, Nlrp3, Fos, Nod1, Myd88, Tnf, C3, Il1a, C1s2, Itgam, C1s1, Il1b, Tlr4, Cxcl5	7.23 × 10^−3^	Casp7, Mapk13, Il6, Irf1	31
mmu05134	Legionellosis	61	0.04	Bnip3, Casp1, Cd14, Tlr2	1.02 × 10^−10^	Pycard, Cycs, Hspa1b, Il12b, Hspa1a, Myd88, Tnf, C3, Hspa8, Itgam, Il1b, Cxcl1, Cxcl2, Tlr4, Hspd1	0.02	Casp7, Il6, Cxcl3	22
mmu05321	Inflammatory bowel disease	62	9.51 × 10^−3^	Il21, Il23r, Il17f, Il23a, Tlr2	1.41 × 10^−7^	Il10, Il1a, Jun, Il12rb2, Il1b, Il13, Il12b, Stat3, Nod2, Tlr4, Tnf, Il17a	3.67 × 10^−4^	Il12rb1, Il22, Il6, Ifng, Stat1	22
mmu04657	IL-17 signaling pathway	91	6.22 × 10^−4^	Ccl11, Il17f, Ikbke, Tnfaip3, S100a9, S100a8, Hsp90aa1, Lcn2	2.57 × 10^−7^	Jun, Il13, Mmp3, Fos, Ccl17, Cebpb, Tnf, Il17a, Il1b, Cxcl1, Csf2, Cxcl2, Csf3, Cxcl5	1.95 × 10^−8^	Mapk13, Il6, Mmp13, Ccl12, Cxcl10, Ccl7, Ifng, Ccl2, Cxcl3, Ptgs2	32
mmu04061	Viral protein interaction with cytokine and cytokine receptor	95	4.35 × 10^−6^	Ccr1, Tnfrsf1b, Ccl11, Ccl22, Ccl9, Cx3cl1, Ccl6, Cxcr1, Ccl3, Ccr8, Ltbr	4.45 × 10^−7^	Il10, Il2ra, Ccl17, Tnf, Tnfsf10, Ccl24, Ccl8, Tnfrsf1a, Ccl4, Xcl1, Cxcl1, Ccr5, Cxcl2, Cxcl5	4.30 × 10^−6^	Cxcl11, Il6, Ccl12, Cxcl10, Cxcl9, Ccl7, Ccl2, Cxcl3	33
mmu04668	TNF signaling pathway	113	9.56 × 10^−3^	Tnfrsf1b, Cx3cl1, Mmp14, Creb3l3, Fas, Creb3l1, Tnfaip3	2.68 × 10^−9^	Jun, Socs3, Mmp3, Lif, Nod2, Fos, Creb5, Cebpb, Tnf, Tnfrsf1a, Ripk3, Il1b, Bcl3, Icam1, Cxcl1, Csf2, Cxcl2, Cxcl5	9.97 × 10^−10^	Casp7, Mapk13, Il6, Ccl12, Cxcl10, Gm5431, Mlkl, Irf1, Ccl2, Ifi47, Cxcl3, Ptgs2	37
mmu05146	Amoebiasis	107	3.97 × 10^−4^	Cd1d2, Col3a1, Arg2, Col1a1, Gnal, Col4a2, Cd14, Serpinb6b, Tlr2	5.31 × 10^−8^	Il10, Arg1, Nos2, Col4a1, Prdx1, Il12b, Tnf, Itgam, Il1b, Serpinb9, Il1r2, Cxcl1, Lamc2, Csf2, Cxcl2, Tlr4	4.24 × 10^−3^	Serpinb9b, Il6, Ifng, Ctsg, Cxcl3	30
mmu05323	Rheumatoid arthritis	87	0.04	Atp6v1a, Ccl3, Ctla4, Il23a, Tlr2	1.44 × 10^−7^	Il11, Jun, Mmp3, Fos, Tnf, Il17a, Il1a, Il1b, Icam1, Cxcl1, Csf2, Cxcl2, Tlr4, Cxcl5	1.72 × 10^−3^	Il6, Ccl12, Ifng, Ccl2, Cxcl3	24
mmu05140	Leishmaniasis	70			7.96 × 10^−9^	Il10, Jun, Fcgr3, Nos2, Il12b, Fos, Myd88, Tnf, C3, Il1a, Itgam, Cybb, Il1b, Tlr4	6.45 × 10^−4^	Fcgr1, Mapk13, Ifng, Stat1, Ptgs2	19
mmu05144	Malaria	57			4.88 × 10^−7^	Il10, Klrb1b, Klrk1, Hgf, Il1b, Icam1, Csf3, Myd88, Tlr4, Tnf, Thbs4	2.46 × 10^−3^	Il6, Ccl12, Ifng, Ccl2	15
mmu05143	African trypanosomiasis	39	8.73 × 10^−3^	Fasl, Fas, Ido2, Ido1	7.65 × 10^−4^	Il10, Il1b, Icam1, Il12b, Myd88, Tnf			10
mmu04625	C-type lectin receptor signaling pathway	112	0.03	Ccl22, Egr2, Card9, Casp1, Il23a, Ikbke	8.09 × 10^−5^	Il10, Clec4d, Jun, Pycard, Fcer1g, Il1b, Bcl3, Clec7a, Il12b, Nlrp3, Ccl17, Tnf	1.21 × 10^−4^	Mapk13, Il6, Irf1, Stat2, Stat1, Clec4e, Ptgs2	25
mmu05167	Kaposi sarcoma-associated herpesvirus infection	225	2.73 × 10^−3^	Ccr1, H2-T24, Hck, H2-K1, H2-Q4, Ccr8, Fas, H2-T3, Ikbke, Gng12, H2-Q1, Tlr3	2.35 × 10^−8^	H2-T23, Jun, H2-T10, Cycs, H2-Q6, H2-Q7, H2-Q2, Fos, Cd200r4, C3, Hif1a, Rcn1, Tnfrsf1a, Gm11127, Myc, Gnb4, Icam1, Stat3, Pik3r6, Cxcl1, Ccr5, Cdkn1a, Csf2, Cxcl2	7.64 × 10^−5^	Mapk13, H2-T22, Il6, Eif2ak2, Irf7, Stat2, Bak1, Stat1, Cxcl3, Ptgs2	46
mmu04145	Phagosome	182	4.35 × 10^−4^	H2-T24, C1rb, Ctss, Lox, Atp6v1a, Lamp2, H2-K1, H2-Q4, Cd14, H2-T3, H2-Q1, Tlr2	1.01 × 10^−6^	H2-T23, C1ra, Fcgr3, H2-T10, Tubb6, H2-Q6, Clec7a, H2-Q7, Tap2, H2-Q2, Itga5, Tuba8, C3, Itgam, Gm11127, Cybb, Olr1, Tlr4, Thbs4	9.58 × 10^−3^	Fcgr1, Msr1, H2-T22, Tubb3, Tap1, Mpo	37
mmu05171	Coronavirus disease—COVID-19	247	2.06 × 10^−3^	C1rb, Sting1, F13a1, Casp1, Ikbke, Hbegf, C2, Selp, C6, Tlr8, Tlr7, Tlr3, Tlr2	3.40 × 10^−8^	C1qb, Mx2, C5ar1, Mx1, Il12b, Tnf, C3, Tnfrsf1a, C3ar1, Ifih1, C1ra, Jun, Mmp3, Nlrp3, Fos, Myd88, C1s2, C1s1, Cybb, Il1b, Stat3, Csf2, Csf3, Tlr4, Cfb	5.61 × 10^−6^	Oas1g, Mapk13, Il6, Ccl12, Cxcl10, Oas2, Cgas, Oas1a, Eif2ak2, Ccl2, Stat2, Stat1	50
mmu00220	Arginine biosynthesis	20			2.22 × 10^−3^	Arg1, Got1, Nos2, Ass1			4
mmu00770	Pantothenate and CoA biosynthesis	20			2.22 × 10^−3^	Aldh1b1, Vnn3, Dpys, Bcat1			4
mmu00524	Neomycin, kanamycin and gentamicin biosynthesis	5					0.05	Hk3	1
mmu05417	Lipid and atherosclerosis	216	5.74 × 10^−3^	Selp, Ero1a, Lox, Fasl, Ccl3, Fas, Casp1, Cd14, Ikbke, Hsp90aa1, Tlr2	1.05 × 10^−8^	Jun, Pycard, Cycs, Mmp3, Hspa1b, Il12b, Hspa1a, Nlrp3, Fos, Myd88, Sod2, Tnf, Tnfsf10, Hspa8, Tnfrsf1a, Cybb, Il1b, Icam1, Stat3, Olr1, Cxcl1, Cxcl2, Tlr4, Hspd1	5.72 × 10^−3^	Casp7, Mapk13, Il6, Ccl12, Irf7, Ccl2, Cxcl3	42
mmu05169	Epstein-Barr virus infection	231	3.38 × 10^−3^	H2-T24, Gadd45g, H2-K1, Gadd45b, H2-Q4, Fas, H2-T3, Ikbke, Tnfaip3, H2-Q1, Cd44, Tlr2	1.07 × 10^−4^	H2-T23, Jun, H2-T10, Cycs, H2-Q6, H2-Q7, Tap2, H2-Q2, Myd88, Tnf, Gm11127, Myc, Icam1, Stat3, Vim, Cdkn1a, Tapbp, B2m	4.16 × 10^−7^	Mapk13, H2-T22, Cxcl10, Eif2ak2, Tap1, Oas1g, Il6, Oas2, Oas1a, Irf7, Stat2, Bak1, Stat1	43
mmu05145	Toxoplasmosis	110			4.83 × 10^−7^	Il10, Nos2, Cycs, Hspa1b, Il12b, Hspa1a, Myd88, Tnf, Hspa8, Tnfrsf1a, Stat3, Pik3r6, Lamc2, Ccr5, Tlr4	4.77 × 10^−3^	Socs1, Mapk13, Ifng, Irgm1, Stat1	20
mmu05142	Chagas disease	103			6.84 × 10^−6^	C1qb, Il10, Jun, Nos2, Il12b, Fos, Myd88, Tnf, C3, Tnfrsf1a, Il1b, Serpine1, Tlr4	3.61 × 10^−3^	Mapk13, Il6, Ccl12, Ifng, Ccl2	18
mmu05164	Influenza A	173			8.65 × 10^−6^	Ifih1, Il33, Pycard, Socs3, Cycs, Mx2, Mx1, Il12b, Nlrp3, Myd88, Tnf, Tnfsf10, Il1a, Tnfrsf1a, Il1b, Icam1, Tlr4	1.37 × 10^−8^	Cxcl10, Eif2ak2, Oas1g, Il6, Ccl12, Ifng, Oas2, Oas1a, Irf7, Ccl2, Stat2, Bak1, Stat1	30
mmu05163	Human cytomegalovirus infection	256	9.67 × 10^−4^	Ccr1, H2-T24, Cx3cl1, Sting1, H2-K1, Creb3l3, H2-Q4, Creb3l1, H2-T3, H2-Q1, Fasl, Ccl3, Fas, Gng12	1.31 × 10^−5^	H2-T23, H2-T10, Cycs, H2-Q6, H2-Q7, Tap2, H2-Q2, Creb5, Tnf, Tnfrsf1a, Gm11127, Il1b, Myc, Ccl4, Gnb4, Stat3, Cdkn2a, Ccr5, Cdkn1a, Tapbp, B2m	9.89 × 10^−4^	Mapk13, H2-T22, Il6, Ccl12, Cgas, Ccl2, Tap1, Bak1, Ptgs2	44
mmu05162	Measles	146			3.92 × 10^−6^	Ifih1, Jun, Cycs, Mx2, Mx1, Hspa1b, Il12b, Il2ra, Hspa1a, Fos, Myd88, Hspa8, Il1a, Il1b, Stat3, Tlr4	1.38 × 10^−5^	Oas1g, Il6, Oas2, Oas1a, Eif2ak2, Irf7, Stat2, Bak1, Stat1	25
mmu04064	NF-kappa B signaling pathway	105	6.47 × 10^−3^	Gadd45g, Gadd45b, Cd14, Ltbr, Tnfaip3, Bcl2a1b, Bcl2a1a	8.17 × 10^−4^	Tnfrsf1a, Plau, Il1b, Ccl4, Icam1, Cxcl1, Cxcl2, Myd88, Tlr4, Tnf			17
mmu05310	Asthma	25			5.17 × 10^−3^	Il10, Fcer1g, Il13, Tnf			4
mmu04217	Necroptosis	174	0.03	Fasl, Fth1, Chmp4b, Fas, Casp1, Tnfaip3, Tlr3, Hsp90aa1	1.44 × 10^−3^	Il33, Pycard, Nlrp3, Pla2g4a, Tnf, Tnfsf10, Il1a, Tnfrsf1a, Ripk3, Cybb, Il1b, Stat3, Tlr4	7.76 × 10^−3^	Zbp1, Ifng, Mlkl, Eif2ak2, Stat2, Stat1	27
mmu05170	Human immunodeficiency virus 1 infection	240	0.01	H2-T24, Tnfrsf1b, Fasl, Sting1, H2-K1, H2-Q4, Fas, H2-T3, Gng12, H2-Q1, Tlr2	1.68 × 10^−5^	H2-T23, Jun, H2-T10, Cycs, H2-Q6, H2-Q7, Tap2, H2-Q2, Fos, Myd88, Tnf, Bst2, Tnfrsf1a, Gm11127, Gnb4, Samhd1, Ccr5, Tapbp, B2m, Tlr4	0.03	Mapk13, H2-T22, Cgas, Trim30d, Tap1, Bak1	37
mmu05161	Hepatitis B	163	7.29 × 10^−3^	Egr2, Fasl, Creb3l3, Fas, Creb3l1, Ikbke, Casp12, Tlr3, Tlr2	7.05 × 10^−3^	Ifih1, Jun, Cycs, Myc, Stat3, Fos, Cdkn1a, Creb5, Myd88, Tlr4, Tnf	0.02	Mapk13, Il6, Irf7, Stat2, Stat1	25
mmu01230	Biosynthesis of amino acids	79			2.16 × 10^−6^	Pkm, Tpi1, Arg1, Got1, Eno1b, Pgk1, Pgam1, Eno1, Gapdh, Bcat1, Pfkp, Ass1			12
mmu05160	Hepatitis C	165			8.80 × 10^−4^	Socs3, Cycs, Ifit1bl1, Mx2, Mx1, Nr1h3, Tnf, Ywhag, Ocln, Tnfrsf1a, Myc, Stat3, Cdkn1a	7.48 × 10^−8^	Oas1g, Cxcl10, Ifng, Oas2, Oas1a, Ifit1bl2, Eif2ak2, Irf7, Stat2, Bak1, Stat1, Ifit1	25
mmu05235	PD-L1 expression and PD-1 checkpoint pathway in cancer	88			3.59 × 10^−3^	Hif1a, Jun, Stat3, Fos, Batf3, Myd88, Tlr4, Batf	1.81 × 10^−3^	Cd274, Mapk13, Ifng, Stat1, Batf2	13
mmu04380	Osteoclast differentiation	128			2.90 × 10^−4^	Il1a, Lilrb4a, Jun, Sirpb1a, Socs3, Fcgr3, Tnfrsf1a, Tnfrsf11b, Il1b, Fos, Tnf, Fosl2	1.71 × 10^−3^	Socs1, Fcgr1, Mapk13, Ifng, Stat2, Stat1	18
mmu05152	Tuberculosis	180			8.51 × 10^−7^	Il10, Fcgr3, Nos2, Vdr, Cycs, Clec7a, Il12b, Nod2, Cebpb, Myd88, Tnf, C3, Il1a, Itgam, Tnfrsf1a, Fcer1g, Il1b, Tlr4, Hspd1	9.10 × 10^−3^	Fcgr1, Mapk13, Il6, Ifng, Stat1, Clec4e	25
mmu04933	AGE-RAGE signaling pathway in diabetic complications	101			2.35 × 10^−3^	Il1a, Jun, Col4a1, Cybb, Il1b, Icam1, Stat3, Serpine1, Tnf	3.31 × 10^−3^	Mapk13, Il6, Ccl12, Ccl2, Stat1	14
mmu04066	HIF-1 signaling pathway	114			7.73 × 10^−7^	Nos2, Eno1b, Timp1, Eno1, Gapdh, Hk2, Hif1a, Ldha, Cybb, Pgk1, Stat3, Serpine1, Cdkn1a, Tlr4, Pfkp			15
mmu04659	Th17 cell differentiation	104			0.03	Hif1a, Jun, Il1b, Il2ra, Stat3, Fos, Il17a	5.80 × 10^−4^	Il12rb1, Il22, Mapk13, Il6, Ifng, Stat1	13
mmu05150	Staphylococcus aureus infection	124			2.29 × 10^−6^	C1qb, Il10, C1ra, Fcgr3, Krt14, C5ar1, Fpr1, C3, C1s2, Itgam, C1s1, Ptafr, C3ar1, Icam1, Cfb			15
mmu05230	Central carbon metabolism in cancer	69			7.45 × 10^−4^	Hif1a, Pkm, Ldha, Myc, Pgam1, Slc16a3, Hk2, Pfkp			8
mmu04658	Th1 and Th2 cell differentiation	88			0.04	Jun, Il12rb2, Il13, Il12b, Il2ra, Fos	0.01	Il12rb1, Mapk13, Ifng, Stat1	10
mmu05203	Viral carcinogenesis	229	8.77 × 10^−3^	H2-T24, Egr2, Scin, H2-K1, Creb3l3, H2-Q4, Ccr8, Creb3l1, H2-T3, Ltbr, H2-Q1	2.36 × 10^−3^	H2-T23, Jun, H2-T10, H2-Q6, H2-Q7, H2-Q2, Creb5, Ywhag, C3, Pkm, Gm11127, Stat3, Cdkn2a, Ccr5, Cdkn1a			26
mmu05165	Human papillomavirus infection	362	0.02	H2-T24, Col1a1, Atp6v1a, Col4a2, H2-K1, Creb3l3, H2-Q4, Creb3l1, H2-T3, Ikbke, H2-Q1, Fasl, Fas, Tlr3	7.78 × 10^−3^	H2-T23, H2-T10, Col4a1, Fzd7, H2-Q6, Mx2, H2-Q7, Mx1, Tnc, H2-Q2, Itga5, Creb5, Tnf, Pkm, Tnfrsf1a, Gm11127, Lamc2, Cdkn1a, Thbs4	0.03	Oasl1, H2-T22, Irf1, Eif2ak2, Stat2, Bak1, Stat1, Ptgs2	41
mmu05135	Yersinia infection	134			1.56 × 10^−3^	Il10, Jun, Pycard, Il1b, Itga5, Nlrp3, Fos, Mefv, Myd88, Tlr4, Tnf	0.04	Mapk13, Il6, Ccl12, Ccl2	15
mmu00330	Arginine and proline metabolism	54			4.24 × 10^−3^	Aldh1b1, Gatm, Arg1, Got1, Nos2, Cndp2			6
mmu04930	Type II diabetes mellitus	48			0.01	Socs3, Pkm, Hpca, Tnf, Hk2			5
mmu05205	Proteoglycans in cancer	205	0.03	Col1a1, Fasl, Sdc4, Fas, Plaur, Met, Cd44, Hbegf, Tlr2	0.01	Hif1a, Plau, Fzd7, Hgf, Myc, Hpse, Il12b, Stat3, Itga5, Cdkn1a, Tlr4, Tnf			21
mmu05418	Fluid shear stress and atherosclerosis	148			9.91 × 10^−3^	Il1a, Jun, Tnfrsf1a, Il1b, Icam1, Il1r2, Fos, Mgst1, Tnf, Ass1	0.02	Txn1, Mapk13, Ccl12, Ifng, Ccl2	15
mmu04216	Ferroptosis	40	9.54 × 10^−3^	Fth1, Slc39a14, Slc39a1, Cp					4
mmu05132	Salmonella infection	253			9.31 × 10^−4^	Jun, Pycard, Tubb6, Cycs, Nlrp3, Fos, Nod1, Gapdh, Myd88, Tnf, Tnfsf10, Tuba8, Tnfrsf1a, Ripk3, Il1b, Myc, Tlr4	0.01	Txn1, Casp7, Mapk13, Il6, Mlkl, Tubb3, Bak1	24
mmu05166	Human T-cell leukemia virus 1 infection	247			7.53 × 10^−6^	H2-T23, Jun, H2-T10, H2-Q6, H2-Q7, Il2ra, H2-Q2, Fos, Creb5, Il15ra, Tnf, Tnfrsf1a, Gm11127, Myc, Icam1, Il1r2, Tspo, Cdkn2a, Cdkn1a, Csf2, B2m			21
mmu01200	Carbon metabolism	122			2.55 × 10^−3^	Pkm, Tpi1, Got1, Eno1b, Pgk1, Pgam1, Eno1, Gapdh, Hk2, Pfkp			10
mmu05210	Colorectal cancer	88			0.01	Jun, Cycs, Myc, Mcub, Fos, Cdkn1a, Ralgds			7
mmu05168	Herpes simplex virus 1 infection	458			0.02	H2-T23, Ifih1, Socs3, H2-T10, Cycs, H2-Q6, H2-Q7, Il12b, Tap2, H2-Q2, Itga5, Myd88, Tnf, Bst2, C3, Tnfrsf1a, Gm11127, Il1b, Tapbp, B2m, Daxx	4.15 × 10^−5^	H2-T22, Eif2ak2, Tap1, Oas1g, Il6, Ccl12, Ifng, Oas2, Cgas, Oas1a, Irf7, Ccl2, Stat2, Bak1, Stat1	36
mmu04664	Fc epsilon RI signaling pathway	66			0.04	Fcer1g, Il13, Pla2g4a, Csf2, Tnf			5
mmu01524	Platinum drug resistance	80			0.03	Slc31a1, Cycs, Cdkn2a, Mgst1, Cdkn1a, Atp7a			6
mmu04978	Mineral absorption	54	0.03	Slc6a19, Fth1, Slc39a1, Steap2					4
mmu04931	Insulin resistance	110			0.01	Ptpn1, Socs3, Tnfrsf1a, Nr1h3, Stat3, Ppargc1b, Creb5, Tnf			8
mmu05231	Choline metabolism in cancer	98			0.02	Hif1a, Jun, Slc44a5, Pdgfc, Pla2g4a, Fos, Ralgds			7
mmu05221	Acute myeloid leukemia	70			0.05	Itgam, Myc, Stat3, Cebpe, Csf2			5
mmu04146	Peroxisome	86			0.04	Prdx5, Nos2, Prdx1, Hp, Sod2, Xdh			6
mmu04932	Non-alcoholic fatty liver disease	151			0.01	Il1a, Jun, Socs3, Tnfrsf1a, Cycs, Il1b, Nr1h3, Cox4i2, Fos, Tnf			10
mmu05222	Small cell lung cancer	93			0.05	Nos2, Col4a1, Cycs, Myc, Lamc2, Cdkn1a			6
mmu00052	Galactose metabolism	32					0.04	Hk3, Mgam	2
mmu04215	Apoptosis—multiple species	32					0.04	Casp7, Bak1	2
mmu04010	MAPK signaling pathway	294			1.91 × 10^−3^	Jun, Cacna1f, Hpca, Hgf, Pdgfc, Mcub, Hspa1b, Hspa1a, Pla2g4a, Fos, Myd88, Tnf, Hspa8, Il1a, Tnfrsf1a, Il1b, Myc, Daxx			18
mmu05322	Systemic lupus erythematosus	148			0.03	C1qb, C3, Il10, C1ra, C1s2, C1s1, Tnf, Elane, Trim21			9
mmu04916	Melanogenesis	100	0.02	AC117663.3, Sco1, Creb3l3, Creb3l1, Mitf, AC110211.1					6
mmu04218	Cellular senescence	184			0.02	H2-T23, Il1a, H2-T10, Gm11127, H2-Q6, Myc, H2-Q7, H2-Q2, Serpine1, Cdkn2a, Cdkn1a			11
mmu05020	Prion disease	268			4.31 × 10^−3^	C1qb, Cacna1f, Tubb6, Cycs, Psmb2, Hspa1b, Hspa1a, Creb5, Tnf, Tuba8, Hspa8, Il1a, Cybb, Il1b, Cox4i2, Psma8			16
mmu00500	Starch and sucrose metabolism	34					0.05	Hk3, Mgam	2
mmu04917	Prolactin signaling pathway	74					6.29 × 10^−3^	Socs1, Mapk13, Irf1, Stat1	4
mmu04142	Lysosome	131	0.02	Slc11a1, Ctss, Ctsd, Npc2, Lamp2, Ctsz, Lgmn					7
mmu04151	PI3K-Akt signaling pathway	359			0.03	Col4a1, Hgf, Pdgfc, Mcub, Il2ra, Tnc, Itga5, Creb5, Ywhag, Myc, Gnb4, Pik3r6, Lamc2, Cdkn1a, Csf3, Tlr4, Thbs4			17
mmu04926	Relaxin signaling pathway	129	0.05	Col3a1, Col1a1, Col4a2, Creb3l3, Creb3l1, Gng12					6
mmu04622	RIG-I-like receptor signaling pathway	70					0.03	Mapk13, Cxcl10, Irf7	3
mmu05200	Pathways in cancer	543			0.04	Jun, Il12rb2, Nos2, Col4a1, Cycs, Fzd7, Hgf, Il13, Il12b, Il2ra, Fos, Il15ra, Ralgds, Csf2rb2, Hif1a, Csf2rb, Myc, Gnb4, Stat3, Cdkn2a, Lamc2, Mgst1, Cdkn1a			23
mmu05202	Transcriptional misregulation in cancer	223	0.05	Gadd45g, Gadd45b, Mitf, Plat, Cd14, Met, Nr4a3, Bcl2a1b, Bcl2a1a					9

^a^ Genes identified from current study involved in stated pathway.

**Table 4 microorganisms-10-00036-t004:** Participation of down-regulated genes in KEGG pathways in the three vaccine strains that protect against ID challenge against SCHU S4 challenge.

KEGG Pathway ID	KEGG Pathway Name	Total Known Genes	Δ*clpB* Alone		Δ*clpB* and LVS Shared		Δ*clpB*, LVS and Δ*gplX* Shared		Total Gene Counts ^a^
			padj	Matched Genes	padj	Matched Genes	padj	Matched Genes	
mmu04080	Neuroactive ligand–receptor interaction	358			2.69 × 10^−5^	Gh, Ghrhr, Chrne, Gabrr2, Npff, Cnr1, Prss2, Lhcgr, Cort, Tshr, Gabra4, Vipr2, Gria2, Gal, Gria1, Sstr2, Chrna6, Grm6, P2rx2, Glp1r, Grik3, Tac2, S1pr5, Htr5b	1.08 × 10^−3^	Grin2a, Npb, Adra1b, Oxtr, Adra2b, Edn3, Ednrb, Npy1r, Aplnr, Gpr156, Vipr1, Ptgfr, Grm4, Gabbr1, Rxfp1, Lpar3	40
mmu04020	Calcium signaling pathway	240			6.53 × 10^−4^	Cacna1g, Atp2b2, Egf, Atp2a3, Fgf18, Plce1, Casq2, Fgfr4, Lhcgr, Plcd3, Camk2b, Pln, P2rx2, Adcy2, Mylk3, Htr5b	1.59 × 10^−4^	Grin2a, Ryr2, Adra1b, Oxtr, Ednrb, Cacna1b, Fgfr3, Fgfr2, Prkcg, Ntrk2, Ptgfr, Camk2a, Ntrk3, Cacna1i	30
mmu05414	Dilated cardiomyopathy	94			8.52 × 10^−4^	Tro, Tnnt2, Pln, Sgca, Atp2a3, Itga8, Adcy2, Sgcg, Ttn	8.30 × 10^−3^	Ryr2, Itga11, Tgfb2, Adcy5, Cacna2d4, Cacng3	15
mmu04512	ECM–receptor interaction	88			8.81 × 10^−3^	Frem2, Vtn, Reln, Tnxb, Col4a3, Col6a6, Itga8	1.27 × 10^−3^	Itga11, Sv2b, Sv2a, Lama3, Col6a4, Fras1, Thbs3	14
mmu05410	Hypertrophic cardiomyopathy	91			2.83 × 10^−3^	Tro, Tnnt2, Sgca, Atp2a3, Prkag3, Itga8, Sgcg, Ttn	7.12 × 10^−3^	Ryr2, Itga11, Ace, Tgfb2, Cacna2d4, Cacng3	14
mmu04916	Melanogenesis	100			0.02	Camk2b, Fzd2, Gnao1, Hr, Adcy2, Wnt2, AC084822.1	0.01	Prkcg, Ednrb, Camk2a, Fzd6, Wnt5b, Adcy5	13
mmu05217	Basal cell carcinoma	63	0.03	Fzd8, Gli1, Wnt10a, Wnt10b			0.03	Bmp4, Apc2, Fzd6, Wnt5b	8
mmu04950	Maturity onset diabetes of the young	27			6.39 × 10^−4^	Bhlha15, Hnf1b, Nr5a2, Foxa2, Foxa3			5
mmu01100	Metabolic pathways	1573	9.01 × 10^−4^	Rimkla, Aldh1a1, Pcx, Selenbp2, Gsta4, Ptdss2, Sec1, B3gnt3, Uros, Mgat3, Suox, Phospho1, Pcyt1b, Cers1, Dhtkd1, B4galnt3, Cox6a2, Hmbs, Mgst3, Hsd3b1, Hagh, Adcy1, Car8, Nags, Mgll, Nqo1, Car2, Gpx1, St3gal5, Pigq, Pik3c2b, Aspdh, Cel, Gck, Cox6b2, Cox8b, Fahd1, Hyal3, Pipox, Urod, Mboat2, Pnpo, Sgpp2, Pip5k1b, Acmsd, Trak2					46
mmu04260	Cardiac muscle contraction	87	1.19 × 10^−3^	Myl4, Actc1, Cox6b2, Cox8b, Cox6a2, Cacng4, Trdn					7
mmu00260	Glycine, serine and threonine metabolism	40			3.94 × 10^−3^	Gamt, Alas2, Cbs, Gcat, Gnmt			5
mmu04310	Wnt signaling pathway	168					4.27 × 10^−3^	Apc2, Prkcg, Tle2, Camk2a, Fzd6, Sox17, Wnt5b, Cxxc4, Dkk2	9
mmu04514	Cell adhesion molecules	174	5.37 × 10^−3^	Cldn13, Cadm3, Cd4, Cdh4, Cldn9, H2-M2, Cd8b1, Nrxn2, Vtcn1					9
mmu04722	Neurotrophin signaling pathway	121					7.58 × 10^−3^	Mapk12, Ntrk2, Camk2a, Ntrk3, Ntf3, Mapk11, Matk	7
mmu05412	Arrhythmogenic right ventricular cardiomyopathy	77			0.02	Actn2, Cdh2, Sgca, Atp2a3, Itga8, Sgcg			6
mmu00860	Porphyrin and chlorophyll metabolism	43	0.05	Hmbs, Uros, Urod					3
mmu04360	Axon guidance	181	0.05	Rac3, Syp, Efna4, Plxnb1, Prkcz, Sema4f, Myl9	0.05	Epha4, Ablim2, Ephb6, Camk2b, Epha8, Hr, Ephb1, Bmp7, Sema4g	6.92 × 10^−3^	Efnb2, Camk2a, Ablim3, Sema6c, Wnt5b, Ntn4, Lrrc4c, L1cam, Rgma	25
mmu04024	cAMP signaling pathway	215			1.58×10^−5^	Atp2b2, Npr1, Hhip, Atp2a3, Plce1, Lhcgr, Cnga2, Tshr, Cnga1, Vipr2, Ppp1r1b, Camk2b, Gria2, Gria1, Pln, Sstr2, Glp1r, Adcy2	0.02	Grin2a, Ryr2, Oxtr, Edn3, Gabbr1, Camk2a, Fxyd1, Npy1r, Adcy5	27
mmu04972	Pancreatic secretion	114	2.77 × 10^−4^	Car2, Cckar, Cela3a, Prss1, Slc12a2, Adcy1, Cel, Slc26a3, Cpa1	0.01	Atp2b2, Atp2a3, Kcnq1, Adcy2, Amy1, Ctrl, Prss2, Cpa2			17
mmu04713	Circadian entrainment	98			0.05	Cacna1g, Camk2b, Gria2, Gria1, Gnao1, Adcy2	8.67 × 10^−5^	Grin2a, Ryr2, Gucy1a1, Prkcg, Camk2a, Gng8, Kcnj9, Adcy5, Cacna1i	15
mmu04974	Protein digestion and absorption	108	0.02	Cela3a, Prss1, Col14a1, Col4a6, Col8a2, Cpa1	5.78×10^−4^	Col11a1, Col4a3, Col13a1, Eln, Col6a6, Kcnq1, Ctrl, Col19a1, Prss2, Cpa2			16
mmu04727	GABAergic synapse	89			2.46×10^−3^	Gabra4, Gls2, Slc12a5, Gabrr2, Gnao1, Slc38a3, Abat, Adcy2	6.40 × 10^−3^	Prkcg, Gabbr1, Gad1, Cacna1b, Gng8, Adcy5	14
mmu04724	Glutamatergic synapse	113			8.26 × 10^−4^	Shank1, Gria2, Gria1, Homer2, Gls2, Grm6, Gnao1, Slc38a3, Adcy2, Grik3	0.02	Grin2a, Pla2g4f, Prkcg, Grm4, Gng8, Adcy5	16
mmu04971	Gastric acid secretion	75			0.01	Atp4a, Camk2b, Sstr2, Kcnq1, Adcy2, Mylk3	2.75 × 10^−3^	Prkcg, Kcnj1, Camk2a, Kcnj16, Adcy5, Kcnf1	12
mmu04911	Insulin secretion	86	0.02	Cckar, Kcnn2, Adcy1, Gcg, Gck			5.42 × 10^−3^	Ryr2, Prkcg, Syt3, Camk2a, Ffar1, Adcy5	11
mmu05200	Pathways in cancer	543	0.05	Nqo1, Fbxo24, Gsta4, Ctnna3, Fzd8, Flt3l, Col4a6, Gli1, Mgst3, Notch3, Gnb3, Rac3, Adcy1, Wnt10a, Hes5, Wnt10b			2.78 × 10^−3^	Apc2, Hlf, Ednrb, Fzd6, Runx1t1, Fgfr3, Fgfr2, Bmp4, Prkcg, Heyl, Tgfb2, Hey2, Camk2a, Lama3, Hey1, Wnt5b, Gng8, Rxrg, Lpar3, Adcy5	36
mmu04925	Aldosterone synthesis and secretion	102			0.05	Cacna1g, Camk2b, Atp2b2, Npr1, Star, Adcy2	2.98 × 10^−3^	Hsd3b6, Kcnk3, Prkcg, Cyp21a1, Camk2a, Adcy5, Cacna1i	13
mmu04261	Adrenergic signaling in cardiomyocytes	152			0.05	Tro, Tnnt2, Camk2b, Pln, Atp2b2, Atp2a3, Kcnq1, Adcy2	7.70 × 10^−3^	Mapk12, Ryr2, Adra1b, Camk2a, Adcy5, Cacna2d4, Cacng3, Mapk11	16
mmu04725	Cholinergic synapse	112			0.03	Ache, Camk2b, Chrna6, Gnao1, Kcnq1, Hr, Adcy2	0.02	Prkcg, Camk2a, Cacna1b, Gng8, Adcy5, Kcnf1	13
mmu05031	Amphetamine addiction	69			0.04	Ppp1r1b, Camk2b, Ddc, Gria2, Gria1	0.04	Grin2a, Prkcg, Camk2a, Adcy5	9
mmu05231	Choline metabolism in cancer	98	0.04	Pcyt1b, Slc22a2, Wasf3, Rac3, Pip5k1b	0.05	Gpcpd1, Egf, Slc22a4, Dgkb, Hr, Chkb			11
mmu04350	TGF-beta signaling pathway	95					1.0 × 10^−45^	Bmp4, Tgfb2, 4930516B21Rik, Nog, Id4, Smad9, Id3, Fmod, Rgma, Thsd4	10
mmu04550	Signaling pathways regulating pluripotency of stem cells	140					1.15 × 10^−5^	Bmp4, Mapk12, Apc2, 4930516B21Rik, Fzd6, Wnt5b, Id4, Smad9, Id3, Fgfr3, Fgfr2, Mapk11	12
mmu04921	Oxytocin signaling pathway	153					1.35 × 10^−4^	Ryr2, Gucy1a1, Pla2g4f, Prkcg, Oxtr, Camk2a, Kcnj9, Adcy5, Kcnf1, Cacna2d4, Cacng3	11
mmu04976	Bile secretion	100	5.54 × 10^−4^	Car2, Kcnn2, Aqp9, Ephx1, Aqp8, Adcy1, Slc22a7, Aqp1					8
mmu04728	Dopaminergic synapse	135					9.55 × 10^−4^	Grin2a, Mapk12, Prkcg, Camk2a, Cacna1b, Gng8, Kcnj9, Adcy5, Mapk11	9
mmu04640	Hematopoietic cell lineage	94	1.87 × 10^−3^	Cd24a, Cd4, Cd59b, Tfrc, Cd8b1, Dntt, Flt3l					7
mmu04934	Cushing syndrome	162					3.36 × 10^−3^	Apc2, Hsd3b6, Kcnk3, Cyp21a1, Camk2a, Fzd6, Wnt5b, Adcy5, Cacna1i	9
mmu04010	MAPK signaling pathway	294					3.48 × 10^−3^	Mapk12, Pla2g4f, Cacna1b, Fgfr3, Cacng3, Fgfr2, Prkcg, Ntrk2, Tgfb2, Ntf3, Cacna1i, Cacna2d4, Mapk11	13
mmu05144	Malaria	57			3.89 × 10^−3^	Gypa, Hbb-bh2, Hba-a1, Hbb-bt, Hbb-bs, Ackr1			6
mmu05033	Nicotine addiction	40			3.94 × 10^−3^	Gabra4, Gria2, Gria1, Chrna6, Gabrr2			5
mmu00350	Tyrosine metabolism	40					6.45 × 10^−3^	Aoc3, Adh1, Aox4, Aox3	4
mmu04723	Retrograde endocannabinoid signaling	148					6.59 × 10^−3^	Mapk12, Prkcg, Faah, Cacna1b, Gng8, Kcnj9, Adcy5, Mapk11	8
mmu05032	Morphine addiction	91					7.12 × 10^−3^	Prkcg, Gabbr1, Cacna1b, Gng8, Kcnj9, Adcy5	6
mmu00514	Other types of O-glycan biosynthesis	43					8.33 × 10^−3^	St6gal1, Colgalt2, Gxylt2, Galnt16	4
mmu00410	beta-Alanine metabolism	31			8.76 × 10^−3^	Upb1, Aldh3a1, Aldh3b2, Abat			4
mmu00750	Vitamin B6 metabolism	9					0.01	Aox4, Aox3	2
mmu04924	Renin secretion	76					0.01	Gucy1a1, Ace, Edn3, Adcy5, Kcnf1	5
mmu04927	Cortisol synthesis and secretion	72					0.01	Hsd3b6, Kcnk3, Cyp21a1, Adcy5, Cacna1i	5
mmu00360	Phenylalanine metabolism	23			0.02	Ddc, Aldh3a1, Aldh3b2			3
mmu00920	Sulfur metabolism	11	0.02	Selenbp2, Suox					2
mmu04977	Vitamin digestion and absorption	24			0.02	Slc23a1, Apoa4, Plb1			3
mmu05143	African trypanosomiasis	39			0.02	Hbb-bh2, Hba-a1, Hbb-bt, Hbb-bs			4
mmu04015	Rap1 signaling pathway	214					0.02	Grin2a, Mapk12, Prkcg, Magi2, Fgfr3, Lpar3, Adcy5, Fgfr2, Mapk11	9
mmu04072	Phospholipase D signaling pathway	149					0.02	Pla2g4f, Ptgfr, Grm4, Dgka, Lpar3, Adcy5, Dnm1	7
mmu04270	Vascular smooth muscle contraction	143					0.02	Gucy1a1, Pla2g2d, Pla2g4f, Prkcg, Adra1b, Edn3, Adcy5	7
mmu04370	VEGF signaling pathway	58					0.02	Mapk12, Pla2g4f, Prkcg, Mapk11	4
mmu04970	Salivary secretion	86					0.02	Gucy1a1, Prkcg, Adra1b, Trpv6, Adcy5	5
mmu05152	Tuberculosis	180					0.02	Cd209g, Mapk12, Cd209f, Tgfb2, Camk2a, Cd209a, Atp6v0a4, Mapk11	8
mmu05418	Fluid shear stress and atherosclerosis	148					0.02	Bmp4, Mapk12, Thbd, 4930516B21Rik, Klf2, Mapk11, Nox1	7
mmu04150	mTOR signaling pathway	156	0.03	Rps6ka6, Stradb, Deptor, Fbxo24, Fzd8, Wnt10a, Wnt10b					7
mmu04014	Ras signaling pathway	232					0.03	Grin2a, Pla2g2d, Pla2g4f, Prkcg, Ntrk2, Ntf3, Gng8, Fgfr3, Fgfr2	9
mmu04330	Notch signaling pathway	60					0.03	Heyl, Tle2, Hey2, Hey1	4
mmu04390	Hippo signaling pathway	157					0.03	Bmp4, Apc2, Tgfb2, Rassf6, Fzd6, Wnt5b, Ajuba	7
mmu04750	Inflammatory mediator regulation of TRP channels	127					0.03	Mapk12, Pla2g4f, Prkcg, Camk2a, Adcy5, Mapk11	6
mmu04912	GnRH signaling pathway	90					0.03	Mapk12, Pla2g4f, Camk2a, Adcy5, Mapk11	5
mmu04913	Ovarian steroidogenesis	63					0.03	Hsd3b6, Pla2g4f, Cyp1a1, Adcy5	4
mmu04926	Relaxin signaling pathway	129					0.03	Mapk12, Ednrb, Gng8, Rxfp1, Adcy5, Mapk11	6
mmu04929	GnRH secretion	63					0.03	Prkcg, Gabbr1, Kcnj9, Cacna1i	4
mmu04960	Aldosterone-regulated sodium reabsorption	38					0.03	Prkcg, Kcnj1, Sfn	3
mmu00910	Nitrogen metabolism	17	0.04	Car2, Car8					2
mmu04710	Circadian rhythm	30			0.04	Npas2, Prkag3, Rorc			3
mmu05135	Yersinia infection	134	0.04	Cd4, Rps6ka6, Fbxo24, Cd8b1, Rac3, Pip5k1b					6
mmu05218	Melanoma	72			0.04	Egf, Fgf18, E2f2, Gadd45a, Hr			5
mmu00250	Alanine, aspartate and glutamate metabolism	39					0.04	Gad1, Aldh5a1, Ddo	3
mmu00760	Nicotinate and nicotinamide metabolism	41					0.04	Aox4, Aox3, Nmnat2	3
mmu00830	Retinol metabolism	97					0.04	Adh1, Aox4, Aox3, Cyp1a1, Lrat	5
mmu00982	Drug metabolism—cytochrome P450	71					0.04	Adh1, Aox4, Aox3, Fmo2	4
mmu04726	Serotonergic synapse	131					0.04	Pla2g4f, Prkcg, Cacna1b, Gng8, Kcnj9, Adcy5	6
mmu04933	AGE-RAGE signaling pathway in diabetic complications	101					0.04	Mapk12, Thbd, Tgfb2, Mapk11, Nox1	5
mmu05205	Proteoglycans in cancer	205					0.04	Mapk12, Prkcg, Tgfb2, Camk2a, Fzd6, Wnt5b, Gpc3, Mapk11	8
mmu00380	Tryptophan metabolism	52			0.05	Afmid, Ddc, Haao, Inmt			4
mmu04918	Thyroid hormone synthesis	74			0.05	Ttr, Adcy2, Duox2, Slc5a5, Tshr			5
mmu05214	Glioma	74			0.05	Camk2b, Egf, E2f2, Gadd45a, Hr			5
mmu05225	Hepatocellular carcinoma	174	0.05	Nqo1, Gsta4, Mgst3, Dpf3, Fzd8, Wnt10a, Wnt10b					7

^a^ Genes identified from current study involved in stated pathway.

**Table 5 microorganisms-10-00036-t005:** Selected biomarkers and their relative expressions and fold changes vs spleens from naïve mice.

Gene Name	Gene Description	Mean					Fold Change			
		Naive	Δ*clpB*	LVS	Δ*gplX*	Δ*lpcC*	Δ*clpB/* Naïve	Δ*clpB/* LVS	*ΔclpB*/ *ΔgplX*	*ΔclpB*/ *ΔlpcC*
*Acod1*	aconitate decarboxylase 1	29	3971	1364	253	31	137.8	2.9	15.7	125.7
*Saa3*	serum amyloid A 3	47	5752	1000	45	21	122.2	5.8	128.1	282.4
*Ccl2*	chemokine (C-C motif) ligand 2	36	1606	685	109	65	44.7	2.3	14.9	24.8
*Clec4e*	C-type lectin domain family 4, member e	83	3073	978	208	105	37.1	3.1	14.8	29.2
*Timp1*	tissue inhibitor of metalloproteinase 1	90	3142	1181	174	95	35.0	2.7	18.1	33.2
*Serpine1*	serine (or cysteine) peptidase inhibitor, clade E, member 1	46	1472	447	70	60	31.8	3.3	21.1	24.7
*Mmp3*	matrix metallopeptidase 3	63	1862	371	111	59	29.8	5.0	16.8	31.7
*Inhba*	inhibin beta-A	34	858	273	36	24	25.2	3.1	24.0	35.1
*Cxcl2*	chemokine (C-X-C motif) ligand 2	13	314	63	12	13	24.5	5.0	25.4	23.9
*Il1rn*	interleukin 1 receptor antagonist	175	4269	1744	389	168	24.4	2.4	11.0	25.4
*Cxcl1*	chemokine (C-X-C motif) ligand 1	29	600	221	45	26	20.8	2.7	13.6	22.8
*Vcan*	versican	45	893	267	95	35	19.9	3.3	9.5	25.6
*Adamts4*	a disintegrin-like and metallopeptidase (reprolysin type) with thrombospondin type 1 motif, 4	23	415	130	33	25	18.2	3.2	12.6	16.9
*Cxcl5*	chemokine (C-X-C motif) ligand 5	54	961	465	96	70	17.8	2.1	10.0	13.9
*Il1r2*	interleukin 1 receptor, type II	45	749	227	87	37	16.7	3.3	8.6	20.3
*Lipg*	lipase, endothelial	42	610	248	90	50	14.5	2.5	6.8	12.2
*Mmp8*	matrix metallopeptidase 8	172	2266	465	314	116	13.2	4.9	7.2	19.6
*Oas1g*	2′-5′ oligoadenylate synthetase 1G	78	884	423	218	86	11.2	2.1	4.1	10.3
*Gzmb*	granzyme B	183	1943	803	497	291	10.6	2.4	3.9	6.7
*Chil1*	chitinase-like 1	96	1003	370	196	90	10.4	2.7	5.1	11.2
*Lox*	lysyl oxidase	95	905	189	47	62	9.5	4.8	19.2	14.6
*Il1a*	interleukin 1 alpha	145	1291	496	164	151	8.9	2.6	7.9	8.5
*Ccl3*	chemokine (C-C motif) ligand 3	75	478	127	83	93	6.3	3.7	5.7	5.2
*Ccl4*	chemokine (C-C motif) ligand 4	102	585	208	130	174	5.7	2.8	4.5	3.4
*Hp*	haptoglobin	654	3737	1587	1119	580	5.7	2.4	3.3	6.4
*Il1b*	interleukin 1 beta	1089	6125	3006	1833	1115	5.6	2.0	3.3	5.5
*Il1f9*	interleukin 1 family, member 9	108	559	227	211	86	5.2	2.5	2.7	6.5
*Aqp1*	aquaporin 1	13520	3451	9079	30537	46957	−3.9	−2.6	−8.9	−13.6
*Sptb*	spectrin beta, erythrocytic	8903	1854	5303	21539	33398	−4.8	−2.9	−11.6	−18.0
*Art4*	ADP-ribosyltransferase 4	406	84	185	610	1184	−4.8	−2.2	−7.3	−14.1
*Slc6a9*	solute carrier family 6 (neurotransmitter transporter, glycine), member 9	648	87	210	1234	2130	−7.5	−2.4	−14.2	−24.6
*1300017J02Rik*	RIKEN cDNA 1300017J02 gene	720	59	170	1114	2409	−12.2	−2.9	−18.9	−40.8

## Data Availability

RNA-seq data are available in the GEO repository with access number GSE186408. The remainder of the data presented in this study are all available in the present article.

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
