# Peer review of "Novel Transcriptional and Translational Biomarkers of Tularemia Vaccine Efficacy in a Mouse Inhalation Model: Proof of Concept"

_microorganisms, 2021, doi:10.3390/microorganisms10010036_

Round 1

Reviewer 1 Report

This study presents a transcriptomics study of mice inoculated with several vaccine candidates for Ft. The results suggest that several biomarkers may be used for vaccine surveillance efforts. I would like to see some additional detail in Table S1 that will allow the reader to understand the variation observed in the transcriptomics count data (see below). Additionally, there are some data results that should be explained and are also noted below:

L122: Where is FSC033 used? If not used, why is it mentioned? Also, LVS should not be a mutant in Table 1, correct?
L154: The definition of naive mice should be explicitly stated in the methods and experimental design
L203: There were only 16 counts for il17a in the clpB mutant. Is that low level of expression expected?
L213: This is all in relation to the naive samples, correct? If so, this info should be added to the figure legend.
L219: I could change the ";" to a ","
L238: Which pathway are you referencing?
L273: I see a Ccl7 in Table S1, not CcL2; are these the same gene?
L276-277: What do you mean by "not sought in this regard"? Please clarify.
L355: How does this list correlate with the results shown in Table S1? There would seem to be other genes that had a higher transcript abundance and fold change compared to naive mice
L388: It is unclear how Table S4 supports this conclusion.

Table S1: Please include the accession numbers for these genes. The p-values, raw and FDR adjusted, should also be added. It would also be nice to show the counts across each replicate so the reader can understand the sample variation. Also, are these counts normalized or raw?
Table 3: What is meant by the total gene counts? Is this how many genes of the pathway that were identified?

Figure 1: Why would LVS_2 cluster with the clpB mutants? The clustering of clpB_1 is also odd compared to other samples and should be addressed in the discussion
Figure 2: What is indicated by the variably colored fonts in this figure?
Figure 3: "s"ignificantly typo in legend. How do the authors explain the one gplX sample with high OD in panel A?

Figure S1: "gentamicin"

Author Response

This study presents a transcriptomics study of mice inoculated with several vaccine candidates for Ft. The results suggest that several biomarkers may be used for vaccine surveillance efforts. I would like to see some additional detail in Table S1 that will allow the reader to understand the variation observed in the transcriptomics count data (see below). Additionally, there are some data results that should be explained and are also noted below:

L122: Where is FSC033 used? If not used, why is it mentioned? Also, LVS should not be a mutant in Table 1, correct?

The reviewer is correct regarding FSC033 and all reference to it has been removed. However, LVS is a spontaneous mutant of a Russian Fth strain and should therefore be included in Table 1. Especially since the title of this table is   “Selected characteristics of F. tularensis strains used in the current study.”

L154The definition of naive mice should be explicitly stated in the methods and experimental design

Already defined in methods at Line 129 in the pdf generated from the original MS. Namely, “Untreated mice were used as negative (naïve) controls.”

L203: There were only 16 counts for il17a in the clpB mutant. Is that low level of expression expected?

Yes and agrees with previous proteomics observations (ref 47). For instance day 4 splenic IL-17 for mice immunized with clpB was 15 ng /ml whilst IFNg and IL-6 were > 1000 ng/ml

L213: This is all in relation to the naive samples, correct? If so, this info should be added to the figure legend.

Done as requested

L219: I could change the ";" to a ","

Done

L238: Which pathway are you referencing

Clarified in revised MS by swapping the term “in this” for  “ in the aforementioned”.

L273: I see a Ccl7 in Table S1, not CcL2; are these the same gene?

In the original Table S1, Ccl7 is found at line 11, and Ccl2 at line 37 are distinct genes

L276-277: What do you mean by "not sought in this regard"? Please clarify.

This statement is redundant to that of the final sentence of the results section and has been removed in the revised MS.

L355: How does this list correlate with the results shown in Table S1? There would seem to be other genes that had a higher transcript abundance and fold change compared to naive mice.

The difference between Table 5 and Table S1 is that the former data has been filtered against the 4 criteria stated in the results line 251-267.  This accounts for the differences between the two tables. To avoid any confusion we have now added the statement “By these criteria, some of the genes ranked highly in Supplementary Table 1, failed to make the grade for inclusion in Table 5.

L388: It is unclear how Table S4 supports this conclusion.

The data in Table S4 shows (line 32 and 33) that Aqp1 and Sptb are both down regulated in clpB and LVS immunized mice, but upregulated in gplx and lpcC vaccinated mice. These results appear to support the conclusion drawn.  

Table S1: Please include the accession numbers for these genes. The p-values, raw and FDR adjusted, should also be added. It would also be nice to show the counts across each replicate so the reader can understand the sample variation.

We added the Accession number to the Ensembl database and p-value and B-H adjusted p-value. With regard to sample variation we provided standard error instead of the read count of each individual replicate sample, which is achievable through the GEO data repository (GSE186408) with a review password (azsrokwahlqvdux).

Also, are these counts normalized or raw?

These are normalised counts as now specified at line 160 in the revised MS

Table 3: What is meant by the total gene counts? Is this how many genes of the pathway that were identified?

Clarified with a footnote in both Tables 3 and 4.

Figure 1:

Heat map shows overall changes, whereas the final potential COP biomarkers needed to pass the statistical criteria stated in the test of the results for Table 5. Variation between individual animals is accommodated by the statistical analyses. Moreover, given that LVS is the second best performing vaccine strain, a certain degree of similarity between transcriptomes of the two strains is expected.

Figure 2: What is indicated by the variably colored fonts in this figure?

All fonts now in un-bolded black and white.

Figure 3: "s"ignificantly typo in legend.

Corrected

How do the authors explain the one gplX sample with high OD in panel A?

The data is the data. The fact is that this sample was not formally recognized as an outlier by ROUT analysis and didn’t affect the interpretation of the data.

Figure S1: "gentamicin"

corrected

Reviewer 2 Report

In the paper tha authors  provide proof-of-concept that unusual host responses to vaccination can potentially serve as novel efficacy biomarkers 
for new tularemia vaccines.

The paper is well written but I have some comments:

Introduction is too long

tables 3 and 4 are too long. Is it possible add the tables as material supplementary?

Add conclusions section and reduce the discussions. 

Author Response

The paper is well written but I have some comments:

Introduction is too long

Introduction has been reduced by 25%

tables 3 and 4 are too long. Is it possible add the tables as material supplementary?

We appreciate that tables 3 and 4 are lengthy compared to more regular data tables. Nevertheless, they are the norm for transcriptomics studies, and given that they are central to this MS, we believe that they should stay within the main text of the results.

Add conclusions section and reduce the discussions. 

A conclusions section has been added, and the discussion reduced by 20%. Also, as a consequence of changes to the introduction and discussion has reduced the number of references by 28.

Round 2

Reviewer 2 Report

The paper can be accepted